# An Integrated Investigation of the Relationship between Two Soil Microbial Communities (Bacteria and Fungi) and *Chrysanthemum Zawadskii (Herb.) Tzvel.* Wilt Disease

**DOI:** 10.3390/microorganisms12020337

**Published:** 2024-02-06

**Authors:** Chao Wu, Juan Peng, Tingting Song

**Affiliations:** Zhejiang Academy of Agricultural Sciences, Hangzhou 310021, China; pengjuan@zaas.ac.cn (J.P.); song_tt@sina.com (T.S.)

**Keywords:** *Chrysanthemum zawadskii (Herb.) Tzvel.*, *Chrysanthemum* wilt, rhizosphere microorganisms, 16S rDNA, community structure

## Abstract

Chrysanthemum wilt is a plant disease that exerts a substantial influence on the cultivation of *Chrysanthemum zawadskii (Herb.)* for tea and beverage production. The rhizosphere microbial population exhibits a direct correlation with the overall health of plants. Therefore, studying the rhizosphere microbial community of *Chrysanthemum zawadskii (Herb.) Tzvel.* is of great significance for finding methods to control this disease. This study obtained rhizosphere soil samples from both diseased and healthy plant individuals and utilized high-throughput sequencing technology to analyze their microbial composition. The results showed that the rhizosphere microbial diversity decreased significantly, and the microbial community structure changed significantly. In the affected soil, the relative abundance of pathogenic microorganisms such as rhizospora and *Phytophthora* was greatly increased, while the relative abundance of beneficial microorganisms such as antagonistic fungi and actinomyces was greatly decreased. In addition, this study also found that soil environmental variables have an important impact on plant resistance; the environmental factors mainly include soil properties, content of major microorganisms, and resistance characteristics of samples. Redundancy analysis showed that the drug-resistant population had a greater impact on the 10 species with the highest abundance, and the environmental factors were more closely related to the sensitive population. In the fungal community, the resistant sample group was more sensitive to the influence of environmental factors and high-abundance fungi. These findings provide a theoretical basis for improving microbial community structure by optimizing fertilization structure, thus affecting the distribution of bacteria and fungi, and thus improving the disease resistance of chrysanthemum. In addition, by regulating and optimizing microbial community structure, new ideas and methods can be provided for the prevention and control of chrysanthemum wilt disease.

## 1. Introduction

*Chrysanthemum zawadskii (Herb.) Tzvel.* (CZHZ) is a prominent variety of chrysanthemum tea in China, cultivated in challenging environmental conditions primarily at an altitude of 800 m. However, with the increasing demand and widespread cultivation of this special type of chrysanthemum tea, the issue of chrysanthemum wilt disease (CWD) has become increasingly important [1]. Despite implementing prevention and control measures, the occurrence of CWD remains difficult to avoid [2]. CWD is a common chrysanthemum disease caused by different pathogens. This disease can cause the leaves and stems of chrysanthemums to gradually wither and rot, ultimately leading to plant death [3]. According to the classification of pathogenic bacteria, CWD can be divided into two types: fungal and bacterial. Fungal CWD is mainly caused by brown rot fungi and gray mold. Brown rot fungi are a collection of filamentous fungi that can cause brown decay of wood, belonging to *Subphylum basidiomycetes*. It has high pathogenicity and may infect poplar, banana, cucumber, and other plants [4,5,6]. It can infect leaves, stems, flower buds, and other parts of chrysanthemums, causing symptoms such as withering, browning, and decay. *Penicillium* and *Pseudomonas* are the major causes of bacterial CWD [7]. *Penicillium* is a common fungus that can spread to chrysanthemums through various means such as soil, seeds, and tools. Once it invades the chrysanthemum tissue, it produces toxins and enzymes that destroy chrysanthemum cells, leading to chrysanthemum wilt. This study demonstrates that CZHZ wilt is caused by fungal CWD, and the main pathogen is *Fusarium oxysporum* f. sp. *chrysanthemi* [8,9]. Zhu Yujing et al. demonstrated that the growth of *Fusarium* is highly susceptible to environmental conditions. The pH level was identified as the primary influencing factor, while light, temperature, and humidity were found to have an impact on the growth of *Fusarium spinosum* [10]. These studies used CZHZ plantation soils to alter the physical and chemical properties that cause CWD. In recent years, we have come to realize that rhizosphere microbial communities (RMCs) play an important role in plant health and disease resistance [11]. The RMC composition of healthy chrysanthemums is stable and diverse, including a variety of fungi, bacteria, archaea, symbiotic bacteria, and probiotics. The levels of pathogenic bacteria, viruses, and nematodes are relatively low. However, the RMC composition and species of diseased chrysanthemum have decreased, and the number of fungi, bacteria, archaea, symbiotic bacteria, and probiotics has decreased significantly. On the contrary, the levels of pathogenic bacteria, viruses, and nematodes are relatively high. These differences may be related to the health status and disease resistance of chrysanthemums. Several studies have found that wilt in edible and medicinal *Chrysanthemum morifolium* leads to a decrease in bacterial richness and diversity. This condition also promotes the growth of *Fusarium* fungi, *Pectobacterium*, and *Dickeya*. Additionally, the proportion of beneficial bacteria in the rhizosphere soil of healthy plants is significantly higher compared to diseased plants [12]. The cutting of *Chrysanthemum morifolium* wilt has a significant effect on the abundance of the bacterial community in rhizosphere soil, and the reduction in species diversity is a key factor in the incidence of chrysanthemum wilt. Limited research has been conducted on the changes in microbial community structure and diversity in the rhizosphere of CZHZ soils affected by disease, particularly regarding the abundance of pathogenic microbial community [13]. In the preliminary study, susceptible and resistant areas of the same farm were identified. A comparison was conducted in this study on the physicochemical parameters and microbial community of soil discovered in the roots of diseased plants, with healthy individuals serving as the control. This study aimed to understand how disease occurrence affects the environmental microecology of chrysanthemums by comparing microbial community and soil physicochemical properties. Our research focus is the rhizosphere microbial community of chrysanthemum plants, and we hope that through this study, we can reasonably control the level of microorganisms to improve the disease resistance of chrysanthemum so as to effectively control the occurrence and spread of chrysanthemum diseases. The research objective was to achieve this goal by using cost-effective methods to regulate the physical and chemical properties of soil. Therefore, the hypothesis proposed in this study is that soil environmental variables have an important impact on plant resistance.

## 2. Materials and Methods

### 2.1. Ethics Statement

All methods were performed in accordance with relevant guidelines and regulations. The samples in this study were collected on a private farm, which is a long-term cooperative institution with the Zhejiang Academy of Agricultural Sciences, and the owners allowed full permission to conduct this study on their sites.

### 2.2. Sample Preparation

The study was carried out in the sampling sites (ca. 100 m × 100 m) in Chun’an, Hangzhou, Zhejiang province, China (geographic coordinate: 30°0′30″ N, 118°54′59″ W). The soil around the roots system of chrysanthemum plants was selected for sampling. First of all, the surface impurities needed to be removed. This specific operation includes the following steps: Prepare some clean basic tools such as shovels, brushes, and scrapers to avoid cross-contamination. Use a shovel or scraper to gently scrape the soil surface of fallen leaves, debris, stones, and other large impurities. Use a brush to gently brush away the small impurities remaining on the surface of the soil, such as small stones, sand, and leaf debris. Check the soil surface meticulously using fingers or a small tool removing impurities and debris, ensuring a spotless surface. It is usually recommended that soil samples are collected at a depth of 10–20 cm to enhance the accuracy and representativeness of the collected samples. The sample fields collected included two healthy plant fields (Hcs) and two fields with symptoms of wilt disease (Scs) (approximately 5 hectares per field). Six chrysanthemums were randomly selected from each Health chrysanthemum (Hc) and each Sick chrysanthemum (Sc). A total of 24 experimental samples were collected. The chrysanthemums in the Hcs were all in a healthy state, while the chrysanthemums in the Scs showed symptoms of wilt disease, such as leaf yellowing, withering, and growth obstruction. The plants in all the fields sampled are of the same age, specifically 6 months old. Furthermore, the soil in all fields is loose, fertile, rich in humus, and well-drained sandy soil. In this study, there are four reasons for selecting the two Hcs and two Scs. First, through direct comparison of the differences in RMCs between healthy plants and plants with blight, microbial community changes related to blight could be more accurately determined. Second, the selection of two Hcs and two Scs can increase the sample diversity of the study. Such a design can improve the reliability and universality of the study and avoid the bias and limitations that a single sample may bring. Third, by selecting an equal number of Hcs and Scs, experimental variables can be more effectively regulated, thus mitigating the impact of extraneous environmental factors on the study’s outcomes. This ensures greater comparability and accuracy of results. Fourth, choosing actual farmland as the research subject can more accurately replicate the authentic growth environment of chrysanthemums, thereby enhancing the practicality of the research findings. This can directly assist and guide agricultural production. Each chrysanthemum sample included a complete root system and sediment tightly attached to each plant. To ensure preservation during shipping, the sample was sealed in an unopened plastic bag and stored in a freezer before being transported to the laboratory. To remove the loosely attached soil, the root was gently shaken. Then, using sterilized fine tweezers and a scalpel, the attached soil was thoroughly scraped off to remove the nodule soil from the fine root. The samples collected from each sampling field were concentrated, homogenized, and stored in a refrigerator at a temperature of −80 °C until further analysis was conducted.

### 2.3. Determination of Soil Chemical Properties

First, the soil and water collected from the sampling site were mixed to generate a soil–water suspension with a ratio of 1:2 (*w*/*v*). The suspension was then agitated at room temperature for 30 min before pH measurement was taken using a glass electrode pH meter (231-01, REX, Shanghai, China). Afterwards, an elemental analyzer vario EL (Elementer, Hanau, Germany) was used to determine the total organic carbon and total organic nitrogen contents in the soil using the dry combustion method. Finally, sodium bicarbonate was used to extract the available phosphorus from the soil, and the content of available phosphorus in the soil was determined using the molybdenum blue method. Five grams each of the initial soil of healthy chrysanthemum plants and diseased chrysanthemum plants was randomly selected for Tukey’s HSD test. The physical and chemical properties of soil from healthy and withered chrysanthemum plants were compared and analyzed based on the test results. Table 1 displays the obtained results, where *p* < 0.05 denotes a significant difference.

### 2.4. DNA Extraction, PCR Amplification, and Construction of Gene Clone Library

After adding liquid nitrogen, the sample was frozen and quickly grounded into fine powder in a sterilized pre-cooled mortar and then transferred to the head tube. For DNA extraction, 0.5 g of each sample was taken. Total DNA extraction was performed according to the instructions of the Power Soil DNA separation kit, and the extracted DNA was stored at −20 °C until subsequent analysis. The amplification and sequencing of the corresponding ribosome-coding genes of bacteria and fungi were completed by Zhejiang Tianke. In bacterial Polymerase Chain Reaction (PCR) amplification, 341F (5-CCTAYGGGRBGCASCAG-3) and 806R (5-ggacannggtatcatat-3) were used as primers to amplify the v3–v4 region of the 16SrRNA gene. The 18SrDNAITS-1 region was sequenced and analyzed for fungal diversity. The primers used in this study were ITS5-1737F (5-GGAGAGTAAAGTCGTAACAA GG-3) and ITS2-2043R (5-gctgcgtcttcatcgc-3), respectively. The amplification conditions were as follows: pre-denaturation at 98 °C for 1 min, followed by denaturation at 98 °C for 10 s, annealing at 50 °C for 30 s, extension at 72 °C for 60 s, 30 cycles, and extension at 72 °C for 5 min. Finally, a gene cloning library was sequenced and constructed using the Illumina HiSeq 2500 platform in Zhejiang Tianke (Hangzhou, China), resulting in 250 bp of end-to-end reads.

### 2.5. Sequence Processing and Analysis

After constructing the gene clone library sequence, FLASH software v1.2.7 was used to merge paired-end readings, and then QIIMEv1.7.0 was used for data processing and UCHIME was used to obtain valid labels. The number of samples used in this sequencing was 12, where a cluster of readings with 97% sequence identity was defined as an operational taxon. Each operational taxonomic unit was annotated using MOTHUR v.1.36.1 and the SSU-rRNA SILVA1.2.8 database [14,15]. QIIMEv1.7.0 was used to sparse data from a monomer-free dataset, a standardized method that filters out sequences unclassified at higher levels as well as non-target sequences (such as mitochondrial and chloroplast DNA) in microbial community data analysis [16] used to eliminate differences in sequencing depth (i.e., number of sequence readings per sample) between different samples when processing microbiome data. By randomly drawing the same number of sequences in each sample, comparability between different samples can be made.

### 2.6. Statistical Analysis

First, we used a vegan software package for statistical analysis of the experimental data. To explore in detail the effects of “location” and “disease” on the sample, we used a one-way analysis of variance to initially assess differences in alpha diversity across groups. On this basis, in order to more accurately detect the effect of multiple factors at the same time, we further implemented a two-factor analysis of variance and analyzed “location” and “disease” as two independent variables. Before performing ANOVA, we verified the normal distribution and homogeneity of variance of the data, which are important prerequisites to ensure the validity of the ANOVA results. Second, the study employed the “heatmap” software available in the R software package to produce heat map images [17]. Venn diagrams were generated using Venn Diagram software [18]. The R software package “ca” was utilized to conduct correspondence analysis to examine changes in bacterial community composition across various samples. The study used LEfSe software (v1.0) to identify the differential abundance of families as biomarkers. Finally, to identify significant differences in the composition of bacterial communities across diverse habitats, we employed PERMANOVA, which was implemented using the adonis function within the vegan software package. Prior to running PERMANOVA, it is crucial to ensure multivariate homogeneity of variance. Therefore, we utilized the betadisper function to check for this assumption and report the results accordingly. To this end, we used the betadisper function, and the detailed results of this verification are presented in Table 1 below.

The test statistics and corresponding *p*-values for each sample group are provided. The *p*-values of each sample group are greater than 0.05, indicating that the assumption of multivariate homogeneity of variance is not violated for that group (Table 1). Based on these results, we can proceed with confidence in applying PERMANOVA to analyze differences in bacterial community composition across habitats. Additionally, in this study, we opted for the Bray–Curtis dissimilarity matrix, which is widely used in microbial ecology studies to assess community composition differences. By incorporating these additional steps and clarifications, we aimed to enhance the robustness and transparency of our statistical analysis.

## 3. Results

### 3.1. Comparison of Initial Soil Physicochemical Properties between Healthy and Diseased Chrysanthemum Plants

The physical and chemical indicators of chrysanthemum withered plants are higher than those of healthy plants (Table 2).

The organic carbon content of healthy chrysanthemum plants is 214.92 ± 27.62 g/kg, lower than that of CWD plants at 426.28 ± 176.23 g/kg. Similarly, the available phosphorus content in healthy chrysanthemum plants is 219.62 ± 51.05 g/kg, which is lower than the CWD individual’s 423.64 ± 202.64 g/kg. Additionally, the soil’s moisture content is 0.49 ± 0.07 g/kg, which is lower than the CWD plant’s 0.70 ± 0.12 g/kg. Furthermore, statistically, this study found no significant difference in total nitrogen, total phosphorus, and soil pH between the two types of chrysanthemums (*p* > 0.05).

### 3.2. Diversity Changes in Rhizosphere Community in Healthy and Diseased Plants

The study found 525,995 high-quality bacterial sequences and 690,488 high-quality fungal sequences through reading and filtering. In addition, each bacterial sample has a high-quality reading range of 30,433 to 36,992 (average length: 408–418 bp). The high-quality reading range for each fungal sample is 22,277 to 54,635 (average length: 208–298 bp).

The number of bacterial operational taxonomic units (OTUs) in the samples of withered plants is 2225.67 ± 235.36, which is significantly higher than that of healthy plants in 1942.33 ± 225.13 (Table 3). The values of bacterial Chao 1 and ACE indicators at the withered samples are 3153.08 ± 317.07 and 3089.63 ± 320.75, respectively, which is significantly higher than those in the healthy samples. The above results indicate that the level of bacterial biodiversity in withered plants is significantly higher than in healthy samples.

A total of 612 fungal OTUs were detected in all libraries, with 533 being present in all samples, 46 being specific to healthy rhizosphere fungi, and 33 being specific to diseased rhizosphere bacteria (Figure 1a). All libraries detected a total of 612 bacterial OTUs (Figure 1b). Of these, 284 were present in all samples, 65 were specific to healthy rhizosphere bacteria, and 81 were specific to diseased rhizosphere bacteria (Figure 1). Subsequently, the study standardized the richness and diversity values of the microbial community of the two groups of samples.

The number of OTUs in healthy chrysanthemum plant samples was the highest. Furthermore, this study revealed that the diversity index of fungi in healthy chrysanthemum plants was comparable to that of wilted chrysanthemum plants, whereas the diversity index of bacteria in wilted chrysanthemum plants exceeded that of healthy chrysanthemum plants (Table 4). There was no difference in fungal community richness between diseased and healthy plants in terms of colony richness.

### 3.3. Classification and Analysis of Rhizosphere Community of Healthy and Diseased Plants

The richness and evenness of the bacterial community within the rhizosphere sample exhibited clear changes, whereas the richness and evenness of the fungal community remained unaltered. Sequencing revealed the diversity of bacterial community in different samples at the phylum level (Figure 2a). In total, 42 phyla were identified, and their relative abundance was determined in all samples, with the first 10 phyla each having a relative abundance greater than 1% (Figure 2a). The phyla of *Proteobacteria*, *Actinobacteria*, *Firmicutes*, *Bacteroidetes*, *Acidobacteria*, *Gemmatimonadetes*, *Chloroflexi*, *Verrucomicrobia*, *Saccharibacteria*, and *Thaumarchaeota* were detected in all samples. However, their relative abundance varied from sample to sample (Figure 2a). *Proteobacteria,* which had the highest relative abundance, exhibited a consistent prevalence in rhizosphere bacteria from both healthy and diseased plants, accounting for over 54.8% of all samples. In rhizosphere, where *Actinomyces* are prevalent, there was a notable decrease in the abundance of diseased bacteria compared to healthy bacteria. In contrast, the abundance of *Firmicutes* and *Gemmatimonadetes* showed significant increases in healthy rhizosphere bacteria (Figure 2a). In addition, the abundance of other dominant phyla such as *Thaumarchaeota*, *Acidobacteria*, and *Bacteroidetes* remained relatively stable in rhizosphere. Sixteen phyla were identified, and the relative abundance of the top ten phyla (each with a relative abundance greater than 1%) in all samples is shown in Figure 1b. The phyla of *Ascomycota, Monoblepharomycota*, *Basidiomycota*, *Chytridiomycota*, *Mortierellomycota*, *Rozellomycota*, *Zoopagomycota*, *Mucoromycota*, *Entomophthoromycota,* and *Blastocladiomycota* were detected in all samples. However, their relative abundances varied across different samples. *Ascomycota* with the highest relative abundance maintained a stable abundance among both healthy and diseased rhizosphere fungi, accounting for more than 53.8% of all samples. However, the relatively high abundance of unknown phyla in the rhizosphere indicated that the infected plants may be influenced by unknown fungal. The abundance of other dominant phyla, such as *Monoblepharomycota*, *Basidiomycota*, and *Mortierellomycota*, remained relatively stable in the rhizosphere. The results of the genus-level sequencing of rhizosphere communities in different samples are depicted in Figure 3.

A total of 614 bacterial genera were identified through sequencing, illustrating the clustering of the top 35 genera. These classified bacterial genera belonged to four phyla (Figure 3a). Among these, 28 genera belonged to *Proteobacteria*, 4 to *Firmicutes*, 1 to *Actinobacteria*, and 1 to *Gemmatimonadetes*. The following genera were the most abundant (>0.2%) across all samples: *Sphingomonas*, *Devosia*, *Rhizomicrobium*, *Pseudolabrys*, *Klebsiella*, *Mizugakiibacter*, *Martelella,* and *Variibacter*. However, the distribution of species varied greatly among the different samples. *Sphingomonas*, *Devosia*, *Rhizomicrobium*, *Pseudolabs*, *Klebsiella, Mizugakiibacter*, *Martellella*, and *Variibacter* were mainly distributed in the pathogenic rhizosphere bacterial microbiota. *Sphingomonas*, *Variibacter*, *Devosia*, *Pseudolabs*, and *Rhizomicrobium* were mainly dominant in the healthy rhizosphere microbiota, with more genera and species of pathogenic rhizosphere microorganisms than healthy ones. *Sphingomonas* and *Devosia* were the dominant genera in the RMCs among all samples, but *Variibacter* had a higher content in healthy rhizosphere samples. A total of 448 fungal genera were identified by sequencing, and these classified fungal genera belong to five phyla (Figure 3b). Among these, 23 genera belonged to *Ascomycota*, 6 to *Basidiomycota*, 2 to *Monoblepharomycota*, 2 to *Chytridiomycota*, and 1 to *Rozellomycota*. After analyzing the healthy samples, it was observed that *Microidum* was among the most prevalent genera present, along with other microorganisms that constituted more than 0.2% of the total population. However, it is noteworthy that the rhizosphere microorganisms, despite being significant, were present in proportions less than 0.2%. However, species distributions differed greatly across different samples. *Didymella, Fusarium, Gibellulopsis, Monoblepharis*, and *Microidium* were mainly distributed in the Sc samples, and *Didymella* and *Fusarium* are important pathogenic bacteria. *Microidium*, *Plectosphaerella*, *Monoblepharis*, *Gibellulopsis,* and *Aspergillus* were dominant in the Hc samples. There was no major fusarium wilt pathogen among them. *Microidum* was the dominant genus in the Hc (37.2%) samples, *Didymella* was the dominant genus in the Sc (13.5%) samples, and *Fusarium* was dominant in the Sc (11.9%) samples, respectively.

### 3.4. LEfSe Analysis of Bacterial and Fungal Community

To analyze the differences in the composition of bacterial and fungal community between the two groups of samples, LEfSe measurements were conducted on the two groups of samples (Figure 4).

At the taxonomic level of bacterial families, it is observed that the abundance of bacteria belonging to *Sphingomonadaceae*, *Xanthobacteraceae*, and *Xanthomonadaceae* is comparatively higher in both samples (Figure 4). In Hc samples, the families *Lachnospiraceae* and *Ruminococcaceae* were significantly enriched, while in Sc samples, *Enterobacteriaceae* and *Caulobacteraceae* were substantially enriched. In addition, PERMANOVA analysis was further performed as described in the Methods section. The results of the PERMANOVA analysis showed that there were significant differences in bacterial community composition between Hc and Sc samples (*p* < 0.05). This result supports our original hypothesis that the bacterial community composition of Hc and Sc samples is different. The ranking map further shows the specific manifestation of this difference, and different sample groups form obvious clusters on the ranking map, indicating that their bacterial community structure is unique. Based on the aforementioned findings, it could be concluded that there was a significant difference in the bacterial community determination results between Hc and Sc samples. *Enterobacteriaceae* and *Caulobacteraceae* were significantly enriched in all Sc samples. Two communities showed significantly different abundances across samples: *Hyphomicrobiaceae* and *Gemmatimonadaceae*. These differentially abundant taxa can be considered potential biomarkers (LDA > 3.0, *p* < 0.05). The bacterial community differed according to the different habitats of *C. zawadskii*. There was a significant difference between the Hc and Sc samples. The results showed that compared to the Hc sample, there were a large number of bacteria in the rhizosphere soil of the Sc sample.

Figure 5 shows the determination results of fungal communities in two sets of samples. The levels of the fungal family, *Plectosphaerellaceae* and *Monoblepharidaceae*, had higher concentrations in all samples, while *Erysiphaceae*, *Aspergillaceae,* and *Erysiphaceae* were found to be significantly enriched in Hc samples (Figure 5). Five families showed significantly different abundances across samples; these differentially abundant taxa can be considered potential biomarkers (LDA > 3.0, *p* < 0.05). The fungal community differed according to the Hc and Sc samples of *C. zawadskii*. The results showed that there were numerous fungi present in the soil of Hc and Sc samples, and significant differences were observed between the two.

### 3.5. Environmental Factors Affecting Soil Microbial Community

To understand the impact of soil environmental variables on plant resistance, based on the analysis of basic soil properties, a comprehensive analysis was conducted on data such as soil properties, main microbial content, and sample resistance characteristics through redundancy analysis.

The interpretation rates of RDA1 and RDA2 in the two soil samples of bacteria were 32.08% and 15.55%, respectively (Figure 6a). Among them, the resistant population has a strong impact on the 10 species of microorganisms with the highest abundance. However, environmental factors are closely related to the sensitive population, indicating that the rich bacterial population can improve the resistance of the resistant population. The percent variations explained of RDA1 and RDA2 in the fungi found in the two soil samples were 20.69% and 14.39%, respectively (Figure 6b). This indicates that in fungi, the resistant population is more sensitive to environmental factors and the impact of a high abundance of fungal midriffs. The positive correlation between bacteria and environmental factors indicates that environmental factors may have a substantial impact on the abundance and species of fungi in soil (Figure 6). In addition, it was also found that the microbial community changes in diseased chrysanthemum plants were larger than those of healthy chrysanthemum plants (Figure 6). The above results indicate that optimizing soil properties and fully understanding the impact of soil environmental variables on plant resistance can effectively improve the microbial community structure in practice. Thus, the distribution of bacteria and fungi can be adjusted, and the disease resistance of chrysanthemums can be improved.

## 4. Discussion

Generally, soil organic carbon serves as both the substrate and metabolite for the energy metabolism and enzyme function of soil microorganisms. Current research shows that the phosphorus content in the affected plant soil increases rapidly, particularly during the decomposition of straw, when phosphorus decomposes at a faster rate [19]. The study found differences in carbon, phosphorus, and water content in diseased soil samples, and a higher organic carbon content in diseased soil compared to healthy soil. This result indicates that the increase in phosphorus may be caused by the decomposition of diseased plant residues, which is similar to the results of Zheng et al. [20]. However, in addition to this, potential changes in the microbial community may also lead to an increase in the rate of phosphorus dissolution. Diseased soils are known to have high levels of organic carbon, which can lead to increased microbial activity. Microbes may release phosphorus in the process of breaking down organic matter, making it available for plants to absorb. At the same time, microorganisms themselves may also change the form and availability of phosphorus in soil through metabolic processes. In addition to the decomposition of plant residues, the increase in phosphorus in soil can be influenced by a variety of other factors, including changes in soil pH, differences in soil texture and structure, and the interaction of other elements in the soil. In addition, the literature has shown that the number of bacteria in nodule soil is significantly higher under high water content conditions than under drought conditions, which is consistent with the results of this experiment [21]. This study found that susceptible plants exhibit significantly higher soil moisture levels compared to healthy plants. Based on previous research, actinomycetes are associated with nutrient cycling, soil quality, crop yield, and plant health [22].

When comparing healthy and diseased soils, there was a significant variation in bacterial population variety and stability. However, in the rhizosphere of *C. zawadskii*, there were fluctuations in the abundance of several taxa (Table 2 and Figure 1). Plant species and root exudates significantly impact the root microbiome community (RMC), resulting in genotype-specific plant communities in identical types of soil [23]. The powdery mildew family is the *Ascomycetes* family. Its species is a parasitic fungus in the aerial parts of higher plants, which contains many plant pathogens and is the main source of toxins that cause plant diseases [24]. The RDA analysis revealed that soil nutrient factors had a more significant effect on microbial composition and that pathogenic bacteria were the main influencing factor in the assembly of the inter-rooted bacterial community. It has been reported that water content, pH, effective nutrients such as phosphorus and nitrogen, and organic matter regulate the assembly of the rhizosphere community and impact the incidence of diseases [25]. In this paper, the LEfSe test results showed significant changes in the abundance of several fungal taxa. Pathogenic fungi of some plant diseases, such as *Didymella*, *Fusarium*, *Gibellulopsis*, *Monoblepharis*, and *Microdium*, were enriched in the rhizosphere. Specifically, *Fusarium*, known for causing chrysanthemum wilting, demonstrated an increased relative abundance in the rhizosphere of chrysanthemums. This observation concurs with the findings presented by Zeng et al., who concluded that *Fusarium* plays a significant role in the pathogenesis of chrysanthemum wilting, and its elevated presence in the rhizosphere is indicative of a potential disease outbreak [26]. The RDA analysis results of this study indicated that soil nutrient factors exert a substantial impact on microbial composition, with pathogenic bacteria being the main influencing factor in the composition of rhizosphere bacterial community. This pertains to a research report indicating that water content, pH value, available nutrient phosphorus, nitrogen, and organic matter are the primary determinants of the rhizosphere community’s composition and their impact on disease occurrence [27]. In addition, this study revealed that the diversity index of Chao1, Shannon, and Simpson of healthy plant roots was higher than that of diseased soil. This is consistent with the findings of Ross et al. [28]. The interaction between soil physicochemical parameters and microbial communities is a complex and dynamic process that significantly influences the overall health of plants. A study in the banana rhizosphere found that soil pH is a fundamental factor that profoundly influences the structure of the microbial community [29]. Many microorganisms exhibit pH preferences, and alterations in soil pH can selectively favor certain microbial groups over others [30]. The availability of essential nutrients, such as nitrogen, phosphorus, and potassium, directly impacts the composition of the microbial community [31]. Nutrient-rich soils can favor the proliferation of certain microbial taxa, while deficiencies can cause changes in community structure [32]. Also, beneficial microorganisms, such as *Actinomyces*, often thrive in soils rich in organic matter, while certain pathogenic fungi can take advantage of those conditions [33]. Likewise, changes in humidity levels can affect microbial metabolism, nutrient diffusion, and interactions [34,35,36]. Soil physical properties, including texture and structure, influencing water retention, aeration, and root penetration [37,38], can determine the suitability of the microbial habitat. For example, compacted soils can impede the movement of beneficial microorganisms while creating favorable niches for certain pathogens [39]. Many of the above studies have shown that healthy plants have higher α diversity than diseased plants, which may be because healthy plants are better adapted to the growing environment, they generally have higher productivity and anti-interference ability, and they have stronger resistance to external interference. This resistance allows them to maintain high species richness and uniformity in the face of disturbance.

In this study, environmental factors, particularly the physical and chemical properties of the soil, had a significant impact on microbial community composition and function. First of all, soil moisture is an important environmental factor, which directly affects the activity and distribution of microorganisms. In our trials, susceptible plants had significantly higher soil moisture than healthy plants, which may have provided more favorable growth conditions for certain microorganisms. The high humidity environment may promote the proliferation of pathogenic bacteria, thus disrupting the balance of the original microbial community, which is consistent with the observed decrease in microbial diversity in the diseased soil [40]. Secondly, soil nutrient content, especially phosphorus, nitrogen, and organic matter content, has a significant impact on microbial community composition. In our study, the increase in phosphorus content was associated with the decay of diseased plant residues, which may have changed the carbon to phosphorus ratio in the soil, which in turn affected microbial metabolism and growth [41]. In addition, nitrogen and organic matter content are also important factors affecting microbial communities. They not only provide essential nutrients for microorganisms but may also indirectly affect microbial communities by affecting physicochemical properties such as soil pH and REDOX potential [42]. In addition to the above factors, soil pH is also a key factor affecting the microbial community [43]. It directly affects the acid–base balance and enzyme activity of microbial cells, thus changing the growth and metabolism of microorganisms. In our study, although the effect of pH was not directly discussed, it is speculated that there may be differences in pH between diseased and healthy soils, which further affect the composition and function of the microbiome.

The results suggested that soil nutrients play a crucial role in the formation of plant rhizosphere communities, which is important in preventing disease invasion. Additionally, the findings highlighted the need to use region-specific biological agents in biological pest control. Region-specific biologics are biologics developed to target pests in specific areas or ecological environments. These agents typically utilize local microbial resources to achieve pest control by enhancing or introducing microorganisms that have a controlling effect on the target pest. Due to differences in the ecological environment and pest species in different regions, the use of region-specific biologics can provide more precise pest control, reduce the use of chemical pesticides, and reduce the impact on the environment and non-target organisms [44,45]. In our results, we found that soil nutrients have an important effect on the formation of plant rhizosphere microbial communities. These microbial communities play a key role in plant growth and development and disease resistance. Further analysis showed that soil nutrient status and microbial community structure were different in different regions, which led to the specificity of plant rhizosphere microbial communities in different regions. This specificity may be closely related to local climate, soil type, vegetation type, and other factors.

This study used sequencing methods to explore the RMC structure, functional diversity, and community structure of chrysanthemums grown under field conditions and elucidated the differences in community structure with plant health status. This study describes the characteristics of microbial diversity associated with both healthy and diseased chrysanthemum plants grown in field conditions, providing valuable insights into the microbial ecology of these plants. Research has demonstrated that there are variations in the microbial composition and functional diversity between healthy and diseased chrysanthemum plant samples grown in the field. *Chrysanthemums* wilt affects bacterial diversity and constructs bacterial communities. The aforementioned findings offer detailed insights into the intricate microbial composition of both healthy chrysanthemum plants and withered chrysanthemum samples cultivated under field conditions. These findings establish a fundamental understanding that can be utilized to enhance the overall wellbeing and development of chrysanthemum plants.

## Figures and Tables

**Figure 1 microorganisms-12-00337-f001:**
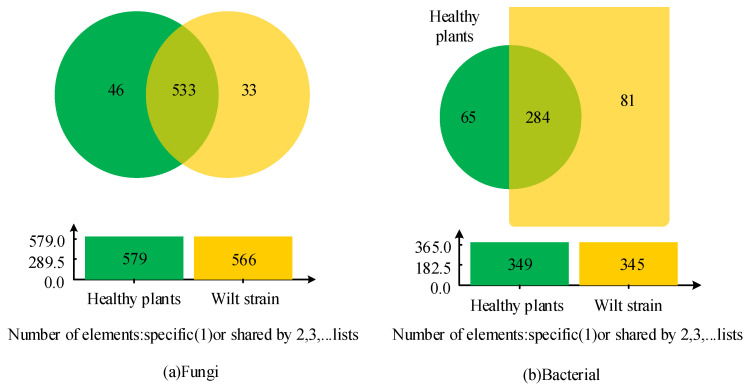
Venn diagrams of unique and common operational taxonomic units for two groups of different samples.

**Figure 2 microorganisms-12-00337-f002:**
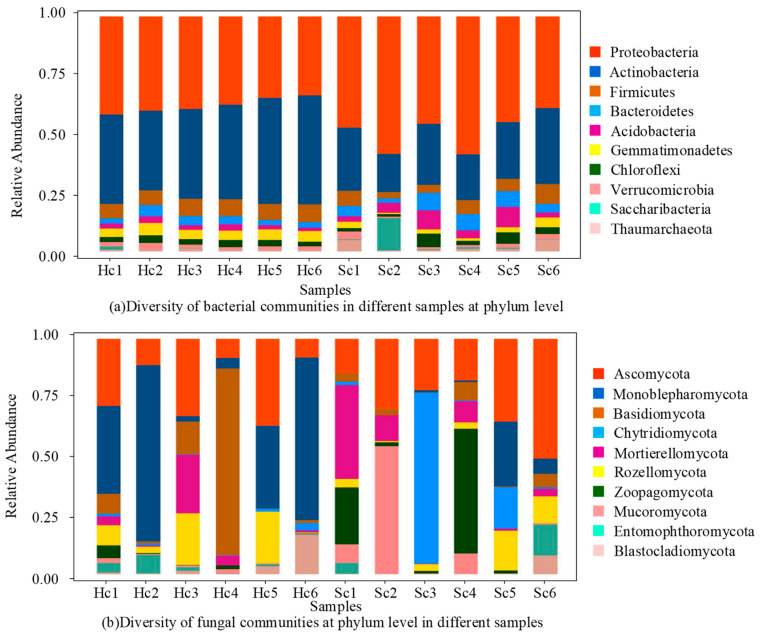
Sequencing results of rhizosphere community of different samples.

**Figure 3 microorganisms-12-00337-f003:**
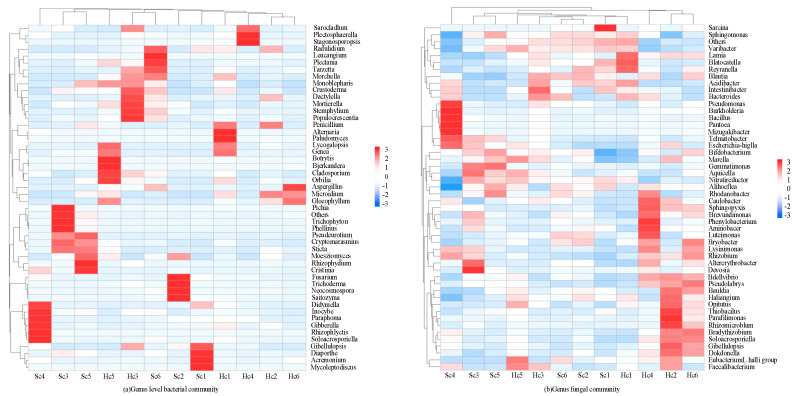
Genus-level sequencing results of rhizosphere communities of different samples. (The data in the figure are z-scores).

**Figure 4 microorganisms-12-00337-f004:**
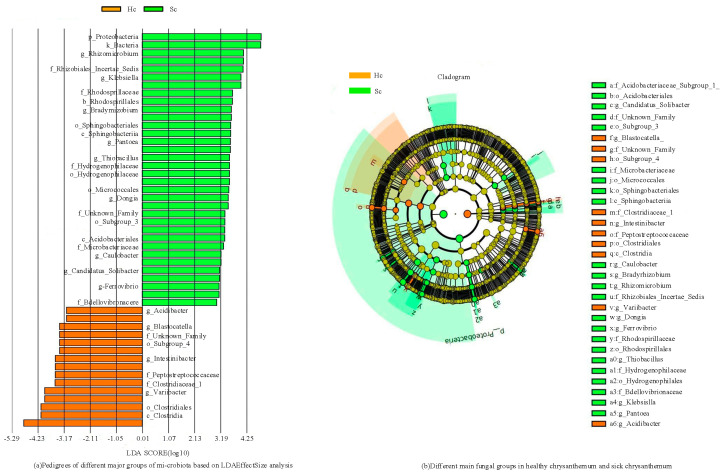
Different main bacterial groups in the two groups of samples.

**Figure 5 microorganisms-12-00337-f005:**
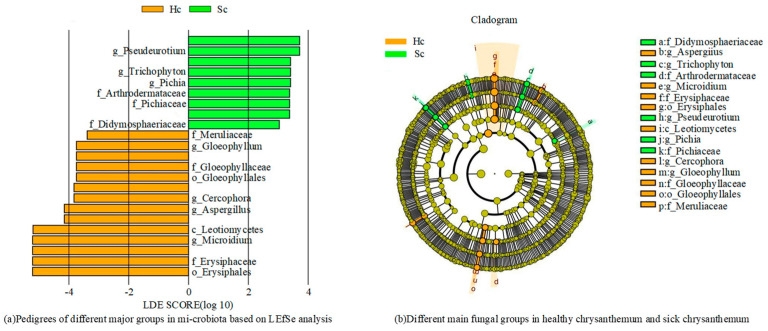
Different main fungal groups in the two groups of samples.

**Figure 6 microorganisms-12-00337-f006:**
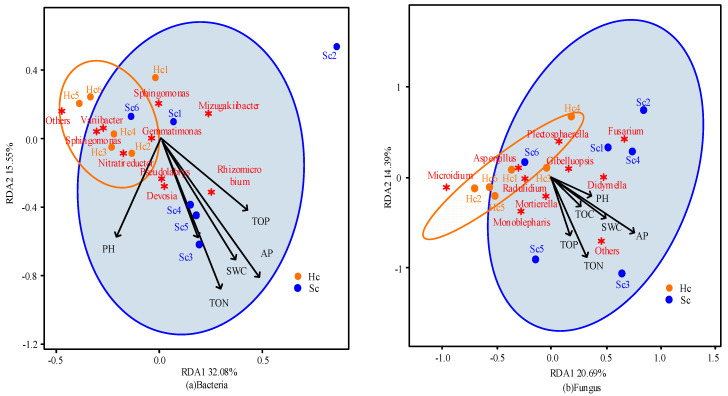
Correlation results between microorganisms and environmental variables in two types of soil samples based on redundancy analysis. The red asterisk (✷) represents top 10 species of microorganisms in bacteria and fungi.

**Table 1 microorganisms-12-00337-t001:** Results of multivariate homogeneity of variance testing using betadisper.

Sample Group	Multivariate Homogeneity of Variance Test Statistic	*p*
Group A	0.132	0.934
Group B	0.258	0.759
Group C	0.334	0.661
Group D	0.469	0.518
Group E	0.618	0.396
Group F	0.757	0.234

**Table 2 microorganisms-12-00337-t002:** Comparison results of soil physicochemical properties between healthy plants and withered plants.

Different Indicators	Healthy Plants	Wilt Strain	*p*
Soil pH value	6.64 ± 0.32	6.56 ± 0.76	1.235
Total nitrogen (g/kg)	5.41 ± 0.62	5.92 ± 1.64	0.078
Total phosphorus (g/kg)	2.05 ± 0.14	2.52 ± 1.15	0.067
Organic carbon (g/kg)	214.92 ± 27.62	426.28 ± 176.23	0.004
Available phosphorus (mg/kg)	219.62 ± 51.05	423.64 ± 202.64	0.003
Water content (g/kg)	0.49 ± 0.07	0.70 ± 0.12	0.005

**Table 3 microorganisms-12-00337-t003:** Diversity of microbial community in different soil treatments. (* indicates *p* < 0.05).

Site	Fungi		Bacterial	
Treatment	*Chrysanthemum* disease plant	*Chrysanthemum* healthy plant	*Chrysanthemum* disease plant	*Chrysanthemum* healthy plant
OTUs observed	365.12 ± 107.34	363.53 ± 107.60	2225.67 ± 235.36 *	1942.33 ± 225.13
Shannon	3.19 ± 1.05	3.49 ± 0.71	8.58 ± 0.31	8.28 ± 0.37
Chao 1	567.32 ± 157.17	543.81 ± 94.61	3153.08 ± 317.07 *	2684.01 ± 331.46
ACE	547.18 ± 164.95	523.33 ± 78.67	3089.63 ± 320.75 *	2636.01 ± 317.01
Coverage (%)	0.99 ± 0.01	0.99 ± 0.01	0.97 ± 0.01	0.97 ± 0.01

**Table 4 microorganisms-12-00337-t004:** Normalized results of each index.

Index	Healthy Plants-b	Wilt Strain-b	Healthy Plants-f	Wilt Strain-f
Shannon	8.75 ± 0.21	8.28 ± 0.37 *	3.29 ± 1.06	3.49 ± 0.71
Simpson	0.991 ± 0.002	0.85 ± 0.007	0.692 ± 0.202	0.768 ± 0.110
Ace	3326.20 ± 193.92	2684.01 ± 331.46 **	622.33 ± 159.46	543.81 ± 94.60
Chao1	3269.05 ± 173.99	2636.01 ± 317.01 **	610.39 ± 163.65	523.33 ± 78.67
Goods_coverage	0.967 ± 0.002	0.974 ± 0.004 **	0.9925 ± 0.0022	0.993 ± 0.001

Note: * indicates significance relative to healthy plants at 0.05. ** indicates significant in 0.01 conditions compared to healthy plants.

## Data Availability

The datasets presented in this study can be found in online repositories. The names of the repository/repositories and accession number(s) can be found below: https://www.ncbi.nlm.nih.gov/sra/PRJNA940325, PRJNA940325 (accessed on 7 March 2023).

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
