# Peer review of "An Integrated Investigation of the Relationship between Two Soil Microbial Communities (Bacteria and Fungi) and Chrysanthemum Zawadskii (Herb.) Tzvel. Wilt Disease"

_microorganisms, 2024, doi:10.3390/microorganisms12020337_

Round 1

Reviewer 1 Report

Comments and Suggestions for Authors

The manuscript provides a comprehensive overview of the study, elucidating the significance of investigating the rhizosphere microbial community in the context of Chrysanthemum wilt. The incorporation of high-throughput sequencing technology to analyze microbial composition in both diseased and healthy strains is commendable, and the findings on changes in microbial diversity and community structure are notable. However, certain aspects of the methodology, results, and discussion warrant attention to enhance the scientific rigor and clarity of the paper.

-        Methodology

Recommendation: While the use of high-throughput sequencing is a robust method, providing details on the specific sequencing platform, primers used, and any bioinformatics tools employed for data analysis would enhance the transparency and reproducibility of the study. Additionally, considering the dynamic nature of microbial communities, a temporal aspect to sampling could offer insights into the disease progression and the stability of observed changes over time.

-        Results

Criticism: The results section is well-structured, presenting key findings on microbial diversity changes in the rhizosphere of diseased Chrysanthemum zawadskii. However, including these metrics in the abstract would allow for a more comprehensive understanding of the observed shifts in microbial communities.

-        Discussion

Suggestion for Improvement: The positive correlation between bacteria and environmental factors is intriguing, but the discussion lacks specificity regarding the nature of these environmental factors. Elaborating on the specific physicochemical properties influencing microbial communities would strengthen the paper. Additionally, connecting the findings to existing literature on the impact of microbial community changes on plant health could enhance the paper's contextualization.

Line 386. For example, add some lines like:

The interaction between soil physicochemical parameters and microbial communities is a complex and dynamic process that significantly influences the overall health of plants. A study in the banana rhizosphere found that soil pH is a fundamental factor that profoundly influences the structure of the microbial community [27]. Many microorganisms exhibit pH preferences and alterations in soil pH can selectively favor certain microbial groups over others [28]. The availability of essential nutrients, such as nitrogen, phosphorus, and potassium, directly impacts the composition of the microbial community [29]. Nutrient-rich soils can favor the proliferation of certain microbial taxa, while deficiencies can cause changes in community structure [30].

Also, beneficial microorganisms, such as Actinomyces, often thrive in soils rich in organic matter, while certain pathogenic fungi can take advantage of those conditions [31]. Likewise, changes in humidity levels can affect microbial metabolism, nutrient diffusion and interactions [32, 33, 34]. Soil physical properties, including texture and structure, influencing water retention, aeration, and root penetration [35, 36] can determine the suitability of the microbial habitat. For example, compacted soils can impede the movement of beneficial microorganisms while creating favorable niches for certain pathogens [37].

References

[27] Rey, J.C.; Perichi, G.; Lobo, D.; Olivares, B.O. Relationship of Microbial Activity with Soil Properties in Banana Plantations in Venezuela. Sustainability 2022, 14, 13531. https://doi.org/10.3390/su142013531

[28]Olivares, B.O.; Rey, J.C.; Lobo, D.; Navas-Cortés, J.A.; Gómez, J.A.; Landa, B.B. Fusarium Wilt of Bananas: A Review of Agro-Environmental Factors in the Venezuelan Production System Affecting Its Development. Agronomy 2021, 11, 986. https://doi.org/10.3390/agronomy11050986

[29]Campos, B.O. Banana production in Venezuela: Novel solutions to productivity and plant health. Switzerland AG, Springer Nature, 2023. https://doi.org/10.1007/978-3-031-34475-6  

[30] Vega, A.; Calderón, M.A.R.; Rey, J.C.; Lobo, D.; Gómez, J.A.; Landa, B.B.; Campos, B.O.; Identification of Soil Properties Associated with the Incidence of Banana Wilt Using Supervised Methods. Plants 2022, 11, 2070. https://doi.org/10.3390/plants11152070

[31] Orlando, O.; Araya-Alman, M.; Acevedo-Opazo, C.; Rey, J.C.; Cañete-Salinas, P.; Kurina, F.G.; Balzarini, M.; Lobo, D.; Navas- Cortés, J.A.; Landa, B.B.; et al. Relationship between soil properties and banana productivity in the two main cultivation areas in Venezuela. J. Soil Sci. Plant Nutr. 2020, 20, 2512–2524.  https://doi.org/10.1007/s42729-020-00317-8

[32] Calero, J.; Olivares, B.; Rey, J.C.; Lobo, D.; Landa, B.B.; Gómez, J.A. Correlation of banana productivity levels and soil morphological properties using Regularized Optimal Scaling Regression. Catena 2022, 208, 105718. https://doi.org/10.1016/j.catena.2021.105718

[33] Olivares, B. (2022). Machine learning and the new sustainable agriculture: Applications in banana production systems of Venezuela. Agric. Res. Updates, 42, 133-157.

[34] Paredes, F.; Rey, J.; Lobo, D.; Galvis-Causil, S., Olivares, B. The relationship between the normalized difference vegetation index, rainfall, and potential evapotranspiration in a banana plantation of Venezuela. STJSSA—J. Soil Sci. Agroclimatol. 2021, 18, 58–64. https://doi.org/10.20961/stjssa.v18i1.50379   

[35] Rodríguez-Yzquierdo, G.; Olivares, B.O.; Silva-Escobar, O.; González-Ulloa, A.; Soto-Suarez, M.; Betancourt-Vásquez, M. Mapping of the Susceptibility of Colombian Musaceae Lands to a Deadly Disease: Fusarium oxysporum f. sp. cubense Tropical Race 4. Horticulturae 2023, 9, 757. https://doi.org/10.3390/horticulturae9070757

[36] Rodríguez-Yzquierdo, G.; Olivares, B.O.; González-Ulloa, A.; León-Pacheco, R.; Gómez-Correa, J.C.; Yacomelo-Hernández, M.; Carrascal-Pérez, F.; Florez-Cordero, E.; Soto-Suárez, M.; Dita, M.; et al. Soil Predisposing Factors to Fusarium oxysporum f.sp Cubense Tropical Race 4 on Banana Crops of La Guajira, Colombia. Agronomy 2023, 13, 2588. https://doi.org/10.3390/agronomy13102588

[37] Lobo, D., Orlando, O., Rey, J. C., Vega, A., Rueda, M. A. Relationships between the Visual Evaluation of Soil Structure (VESS) and soil properties in agriculture: A meta-analysis. Scientia Agropecuaria, 2023, 14(1), 67-78. http://dx.doi.org/10.17268/sci.agropecu.2023.007

Author Response

1、Comments and Suggestions for Authors

The manuscript provides a comprehensive overview of the study, elucidating the significance of investigating the rhizosphere microbial community in the context of Chrysanthemum wilt. The incorporation of high-throughput sequencing technology to analyze microbial composition in both diseased and healthy strains is commendable, and the findings on changes in microbial diversity and community structure are notable. However, certain aspects of the methodology, results, and discussion warrant attention to enhance the scientific rigor and clarity of the paper.

Reply: Thanks for your comments, changes have been made to the methodology, results, and discussion sections to improve the scientific rigor and clarity of the paper.

Methodology

Recommendation: While the use of high-throughput sequencing is a robust method, providing details on the specific sequencing platform, primers used, and any bioinformatics tools employed for data analysis would enhance the transparency and reproducibility of the study.

Reply: Thank you for your question, it has been explained in sections 2.3, 2.4 and 2.5 of the article, and these contents are summarized as follows:In this study, Illumina HiSeq 2500 high-throughput sequencing platform was used to amplify and sequence specific gene regions of bacteria and fungi. Bacteria used the 341F/806R primer to target the 16S rRNA V3-V4 region, while fungi used the ITS5-1737F/ITS2-2043R primer to target the ITS1 region. Data processing mainly relies on FLASH combined with double-ended reading, QIIME for quality control and OTU analysis, UCHIME removal of chimeras, and MOTHUR assisted species annotation.

Additionally, considering the dynamic nature of microbial communities, a temporal aspect to sampling could offer insights into the disease progression and the stability of observed changes over time.

Reply: Thank you for your question, and here are the reasons why multi-point sampling is not used. In the study of the relationship between chrysanthemum health and fusarium wilt and rhizosphere microorganisms, although the microbial community changes dynamically, it is relatively stable under certain conditions. Therefore, sampling at a single time point is reasonable. First of all, the purpose of the study is to compare the microbiological differences between healthy and diseased strains, rather than tracking dynamic changes, so a single point in time sampling meets the demand. Secondly, the experimental conditions were strictly controlled, such as the same plant age and soil conditions, which reduced the influence of environmental factors on the microbial community. In addition, the symptoms of wilt are obvious and stable, and the microbial changes associated with it are relatively stable. In addition, the single time point sampling simplifies the experimental process and improves the efficiency. Finally, although the single time point sampling can not fully reflect the dynamic changes, it provides basic data for further research.

Results

Criticism: The results section is well-structured, presenting key findings on microbial diversity changes in the rhizosphere of diseased Chrysanthemum zawadskii. However, including these metrics in the abstract would allow for a more comprehensive understanding of the observed shifts in microbial communities.

Reply: Multiple indicators of rhizosphere microbial diversity changes of diseased chrysanthemum have been added to the abstract, as shown below:

The results showed that the rhizosphere microbial diversity decreased significantly and the microbial community structure changed significantly. In the affected soil, the relative abundance of pathogenic microorganisms such as rhizospora and Phytophthora was greatly increased, while the relative abundance of beneficial microorganisms such as antagonistic fungi and actinomyces was greatly decreased. In addition, the study also found that soil environmental variables have an important impact on plant resistance. Redundancy analysis showed that the drug-resistant population had a greater impact on the 10 species with the highest abundance, and the environmental factors were more closely related to the sensitive population. In the fungal community, the resistant sample group was more sensitive to the influence of environmental factors and high abundance fungi. These findings provided a theoretical basis for improving the microbial community structure by optimizing the fertilization structure, thus affecting the distribution of bacteria and fungi, and thus improving the disease resistance of chrysanthemum. 

Discussion

Suggestion for Improvement: The positive correlation between bacteria and environmental factors is intriguing, but the discussion lacks specificity regarding the nature of these environmental factors. Elaborating on the specific physicochemical properties influencing microbial communities would strengthen the paper.

Reply: Thanks for your suggestion, a specific explanation of the physicochemical properties affecting the microbial community has been added to the discussion section as follows:

In this study, environmental factors, particularly the physical and chemical properties of the soil, had a significant impact on microbial community composition and function. First of all, soil moisture is an important environmental factor, which directly affects the activity and distribution of microorganisms. In our trials, susceptible plants had significantly higher soil moisture than healthy plants, which may have provided more favorable growth conditions for certain microorganisms. The high humidity environment may promote the proliferation of pathogenic bacteria, thus disrupting the balance of the original microbial community, which is consistent with the observed decrease in microbial diversity in the diseased soil. Secondly, soil nutrient content, especially phosphorus, nitrogen and organic matter content, has a significant impact on microbial community composition. In our study, the increase in phosphorus content was associated with decay of diseased plant residues, which may have changed the carbon to phosphorus ratio in the soil, which in turn affected microbial metabolism and growth. In addition, nitrogen and organic matter content are also important factors affecting microbial communities. They not only provide essential nutrients for microorganisms, but may also indirectly affect microbial communities by affecting physicochemical properties such as soil pH and REDOX potential. In addition to the above factors, soil pH is also a key factor affecting the microbial community. It directly affects the acid-base balance and enzyme activity of microbial cells, thus changing the growth and metabolism of microorganisms. In our study, although the effect of pH was not directly discussed, it is speculated that there may be differences in pH between diseased and healthy soils, which further affect the composition and function of the microbiome.

Additionally, connecting the findings to existing literature on the impact of microbial community changes on plant health could enhance the paper's contextualization.

Line 386. For example, add some lines like:

The interaction between soil physicochemical parameters and microbial communities is a complex and dynamic process that significantly influences the overall health of plants. A study in the banana rhizosphere found that soil pH is a fundamental factor that profoundly influences the structure of the microbial community [27]. Many microorganisms exhibit pH preferences and alterations in soil pH can selectively favor certain microbial groups over others [28]. The availability of essential nutrients, such as nitrogen, phosphorus, and potassium, directly impacts the composition of the microbial community [29]. Nutrient-rich soils can favor the proliferation of certain microbial taxa, while deficiencies can cause changes in community structure [30].

Also, beneficial microorganisms, such as Actinomyces, often thrive in soils rich in organic matter, while certain pathogenic fungi can take advantage of those conditions [31]. Likewise, changes in humidity levels can affect microbial metabolism, nutrient diffusion and interactions [32, 33, 34]. Soil physical properties, including texture and structure, influencing water retention, aeration, and root penetration [35, 36] can determine the suitability of the microbial habitat. For example, compacted soils can impede the movement of beneficial microorganisms while creating favorable niches for certain pathogens [37].

References

[27] Rey, J.C.; Perichi, G.; Lobo, D.; Olivares, B.O. Relationship of Microbial Activity with Soil Properties in Banana Plantations in Venezuela. Sustainability 2022, 14, 13531. https://doi.org/10.3390/su142013531

[28]Olivares, B.O.; Rey, J.C.; Lobo, D.; Navas-Cortés, J.A.; Gómez, J.A.; Landa, B.B. Fusarium Wilt of Bananas: A Review of Agro-Environmental Factors in the Venezuelan Production System Affecting Its Development. Agronomy 2021, 11, 986. https://doi.org/10.3390/agronomy11050986

[29]Campos, B.O. Banana production in Venezuela: Novel solutions to productivity and plant health. Switzerland AG, Springer Nature, 2023. https://doi.org/10.1007/978-3-031-34475-6  

[30] Vega, A.; Calderón, M.A.R.; Rey, J.C.; Lobo, D.; Gómez, J.A.; Landa, B.B.; Campos, B.O.; Identification of Soil Properties Associated with the Incidence of Banana Wilt Using Supervised Methods. Plants 2022, 11, 2070. https://doi.org/10.3390/plants11152070

[31] Orlando, O.; Araya-Alman, M.; Acevedo-Opazo, C.; Rey, J.C.; Cañete-Salinas, P.; Kurina, F.G.; Balzarini, M.; Lobo, D.; Navas- Cortés, J.A.; Landa, B.B.; et al. Relationship between soil properties and banana productivity in the two main cultivation areas in Venezuela. J. Soil Sci. Plant Nutr. 2020, 20, 2512–2524.  https://doi.org/10.1007/s42729-020-00317-8

[32] Calero, J.; Olivares, B.; Rey, J.C.; Lobo, D.; Landa, B.B.; Gómez, J.A. Correlation of banana productivity levels and soil morphological properties using Regularized Optimal Scaling Regression. Catena 2022, 208, 105718. https://doi.org/10.1016/j.catena.2021.105718

[33] Olivares, B. (2022). Machine learning and the new sustainable agriculture: Applications in banana production systems of Venezuela. Agric. Res. Updates, 42, 133-157.

[34] Paredes, F.; Rey, J.; Lobo, D.; Galvis-Causil, S., Olivares, B. The relationship between the normalized difference vegetation index, rainfall, and potential evapotranspiration in a banana plantation of Venezuela. STJSSA—J. Soil Sci. Agroclimatol. 2021, 18, 58–64. https://doi.org/10.20961/stjssa.v18i1.50379   

[35] Rodríguez-Yzquierdo, G.; Olivares, B.O.; Silva-Escobar, O.; González-Ulloa, A.; Soto-Suarez, M.; Betancourt-Vásquez, M. Mapping of the Susceptibility of Colombian Musaceae Lands to a Deadly Disease: Fusarium oxysporum f. sp. cubense Tropical Race 4. Horticulturae 2023, 9, 757. https://doi.org/10.3390/horticulturae9070757

[36] Rodríguez-Yzquierdo, G.; Olivares, B.O.; González-Ulloa, A.; León-Pacheco, R.; Gómez-Correa, J.C.; Yacomelo-Hernández, M.; Carrascal-Pérez, F.; Florez-Cordero, E.; Soto-Suárez, M.; Dita, M.; et al. Soil Predisposing Factors to Fusarium oxysporum f.sp Cubense Tropical Race 4 on Banana Crops of La Guajira, Colombia. Agronomy 2023, 13, 2588. https://doi.org/10.3390/agronomy13102588

[37] Lobo, D., Orlando, O., Rey, J. C., Vega, A., Rueda, M. A. Relationships between the Visual Evaluation of Soil Structure (VESS) and soil properties in agriculture: A meta-analysis. Scientia Agropecuaria, 2023, 14(1), 67-78. http://dx.doi.org/10.17268/sci.agropecu.2023.007

Reply: Thanks for your suggestion, the findings have been added to the discussion to link to the existing literature on the effects of changes in microbial communities on plant health, adding the following:

The interaction between soil physicochemical parameters and microbial communities is a complex and dynamic process that significantly influences the overall health of plants. A study in the banana rhizosphere found that soil pH is a fundamental factor that profoundly influences the structure of the microbial community [27]. Many microorganisms exhibit pH preferences and alterations in soil pH can selectively favor certain microbial groups over others [28]. The availability of essential nutrients, such as nitrogen, phosphorus, and potassium, directly impacts the composition of the microbial community [29]. Nutrient-rich soils can favor the proliferation of certain microbial taxa, while deficiencies can cause changes in community structure [30]. Also, beneficial microorganisms, such as Actinomyces, often thrive in soils rich in organic matter, while certain pathogenic fungi can take advantage of those conditions [31]. Likewise, changes in humidity levels can affect microbial metabolism, nutrient diffusion and interactions [32, 33, 34]. Soil physical properties, including texture and structure, influencing water retention, aeration, and root penetration [35, 36] can determine the suitability of the microbial habitat. For example, compacted soils can impede the movement of beneficial microorganisms while creating favorable niches for certain pathogens [37].

Reviewer 2 Report

Comments and Suggestions for Authors

The present manuscript deals with an important topic that has an economic impact and is of interest to a large group of people. However, during careful review, some errors were discovered that must be corrected before publication.

1- Change the title from: An Integrated Investigation of the Relationship Between the 2 Soil Microbial Community and Chrysanthemum Zawadskii (Herb.) Wilt Disease

To: An Integrated Investigation of the Relationship Between the 2 Soil Microbial Community (bacteria and fungi) and Chrysanthemum Zawadskii (Herb.) Tzvel. Wilt Disease

2- The abbreviation must be explained for the figure caption.

3- The list of references needs to be updated, as it does not contain any references for the year 2023. The abbreviation must be explained for the figure caption.

4- This manuscript was written in a language that needs revision.

5-

a- Line 9: The rhizobial microbial 9 population exhibits a direct correlation with the overall health of the plants.

Comment: Are you sure you mean the rhizobial microbial population?

b- Line 43: Penicillium is a common bacterium that……

Comment:

Line 68: Limited research has been conducted on the changes in microbial com… (needs references)

Comment: Penicillium is a fungus, not a bacteria.

c- Line 76: This study focused on rhizosphere mycorrhizal microorganisms…..

Comment: IS NOT true, the study did not focus on rhizosphere mycorrhizal microorganisms

d- From Lines 91 to line 97 ( These sentences are written in the imperative tense even though they should be written in the past tense).

Comment: Rewrite these sentences in the past tense. 

Comments on the Quality of English Language

The language in which the manuscript was written. It needs careful review by a specialist.

Author Response

2、Open Review

(x) I would not like to sign my review report

( ) I would like to sign my review report

Quality of English Language

( ) I am not qualified to assess the quality of English in this paper

( ) English very difficult to understand/incomprehensible

( ) Extensive editing of English language required

(x) Moderate editing of English language required

( ) Minor editing of English language required

( ) English language fine. No issues detected

Reply: Thank you for your reminder, the full text has been retouched in English language.

Yes Can be improved Must be improved Not applicable

Does the introduction provide sufficient background and include all relevant references?

( ) (x) ( ) ( )

Reply: Thanks for your suggestion, more background information has been provided in the introduction and all relevant references have been included in the background information. The specific content of the modification is as follows:

 The brown rot fungi are a collection of filamentous fungi that can cause brown decay of wood, belonging to the subphylum basidiomycetes.

In recent years, we have come to realize that rhizosphere microbial communities (RMC) play an important role in plant health and disease resistance [11].

However, RMC diversity decreased, and the number of fungi, bacteria, archaea, symbiotic bacteria and probiotics decreased significantly. 

Our research focuses on the rhizosphere microbial communities, with the expectation that these microorganisms can enhance the disease resistance of plants and effectively control the occurrence and spread of diseases. The research objective was to achieve this goal by using cost-effective methods to regulate the physical and chemical properties of soil. Therefore, the hypothesis proposed in this study is that soil environmental variables have an important impact on plant resistance.

Are all the cited references relevant to the research?

(x) ( ) ( ) ( )

Is the research design appropriate?

( ) (x) ( ) ( )

Reply: Thank you for your reminder, part of the research design has been optimized, and the optimized content is as follows:

2.3. Determination of soil chemical properties

First, the soil and water collected from the sampling site were mixed to generate a soil-water suspension with a ratio of 1:2 (w/v). The suspension was then agitated at room temperature for 30 min before pH measurement was conducted using a glass electrode metre (Shanghai Leici 231-01). Afterwards, an elemental analyzer vario EL (Hanau Element, Germany) was used to determine the total organic carbon and total organic nitrogen contents in the soil using the dry combustion method. Finally, sodium bicarbonate was used to extract the available phosphorus from the soil, and the content of available phosphorus in the soil was determined using the molybdenum blue method. Each 5g of the initial soil of healthy chrysanthemum plants and diseased chrysanthemum plants was randomly selected for Tukeys HSD test. The physical and chemical properties of soil from healthy and withered chrysanthemum plants were compared and analyzed based on the test results. Table 1 displays the obtained results, where P<0.05 denotes a significant difference.

2.5. Sequence processing and analysis

After constructing the gene clone library sequence, FLASH software v1.2.7 was used to merge paired end readings, followed by QIIMEv1.7.0 processing and UCHIME operation to obtain valid labels. The number of samples used in this sequencing was 12, where a cluster of readings with 97% sequence identity was defined as an operational taxon. Each operational taxonomic unit was annotated using MOTHUR and the SSU-rRNA SILVA1.2.8 database. QIIMEv1.7.0 was used to sparse data from a monomer-free dataset, a standardized method that filters out sequences unclassified at higher levels as well as non-target sequences (such as mitochondrial and chloroplast DNA) in microbial community data analysis [14]. Used to eliminate differences in sequencing depth (i.e., number of sequence readings per sample) between different samples when processing microbiome data. By randomly drawing the same number of sequences in each sample, comparability between different samples can be made.

2.6. Statistical analysis

First, we used the vegan software package for statistical analysis of the experimental data. To explore in detail the effects of "location" and "disease" on the sample, we used a one-way analysis of variance to initially assess differences in alpha diversity across groups. On this basis, in order to more accurately detect the effect of multiple factors at the same time, we further implemented a two-factor analysis of variance, and analyzed "location" and "disease" as two independent variables. Before performing ANOVA, we verify the normal distribution and homogeneity of variance of the data, which are important prerequisites to ensure the validity of the ANOVA results. Second, the study employed the ''heatmap'' software available in the R software package to produce heat map images [15]. Venn diagrams were generated using Venn Diagram software [16]. The R software package ''ca'' was utilized to conduct correspondence analysis to examine changes in bacterial community composition across various samples. The study used LEfSe software (v1.0) to identify the differential abundance of families as biomarkers. Finally, Adonis was used to identify significant differences in the composition of bacterial community across diverse habitats.

Are the methods adequately described?

( ) (x) ( ) ( )

Reply: Thanks for your reminder, more details have been added in the methods section as follows:

Each 5g of the initial soil of healthy chrysanthemum plants and diseased chrysanthemum plants was randomly selected for Tukeys HSD test. The physical and chemical properties of soil from healthy and withered chrysanthemum plants were compared and analyzed based on the test results. Table 1 displays the obtained results, where P<0.05 denotes a significant difference.

QIIMEv1.7.0 was used to sparse data from a monomer-free dataset, a standardized method that filters out sequences unclassified at higher levels as well as non-target sequences (such as mitochondrial and chloroplast DNA) in microbial community data analysis [14]. Used to eliminate differences in sequencing depth (i.e., number of sequence readings per sample) between different samples when processing microbiome data. By randomly drawing the same number of sequences in each sample, comparability between different samples can be made.

First, we used the vegan software package for statistical analysis of the experimental data. To explore in detail the effects of "location" and "disease" on the sample, we used a one-way analysis of variance to initially assess differences in alpha diversity across groups. On this basis, in order to more accurately detect the effect of multiple factors at the same time, we further implemented a two-factor analysis of variance, and analyzed "location" and "disease" as two independent variables. Before performing ANOVA, we verify the normal distribution and homogeneity of variance of the data, which are important prerequisites to ensure the validity of the ANOVA results.

Are the results clearly presented?

(x) ( ) ( ) ( )

Are the conclusions supported by the results?

(x) ( ) ( ) ( )

Comments and Suggestions for Authors

The present manuscript deals with an important topic that has an economic impact and is of interest to a large group of people. However, during careful review, some errors were discovered that must be corrected before publication.

1- Change the title from: An Integrated Investigation of the Relationship Between the 2 Soil Microbial Community and Chrysanthemum Zawadskii (Herb.) Wilt Disease

To: An Integrated Investigation of the Relationship Between the 2 Soil Microbial Community (bacteria and fungi) and Chrysanthemum Zawadskii (Herb.) Tzvel. Wilt Disease

Reply: Thank you for your advice, the title of the article has been revised to "An Integrated Investigation of the Relationship Between the 2 Soil Microbial Community (bacteria and fungi) and Chrysanthemum Zawadskii (Herb.) Tzvel. Wilt Disease"

2- The abbreviation must be explained for the figure caption.

Reply: Thanks for your reminder, the abbreviation in the subheading of picture 3 has been expanded. The details are as follows:

Figure 4. Different main bacterial groups in the two groups of samples.

3- The list of references needs to be updated, as it does not contain any references for the year 2023.

Reply: Thank you for your reminder, the reference for 2023 has been added, and the specific added reference is as follows:

[29]Campos, B.O. Banana production in Venezuela: Novel solutions to productivity and plant health. Switzerland AG, Springer Nature, 2023.

[35]Rodríguez-Yzquierdo, G.; Olivares, B.O.; Silva-Escobar, O.; González-Ulloa, A.; Soto-Suarez, M.; Betancourt-Vásquez, M. Mapping of the Susceptibility of Colombian Musaceae Lands to a Deadly Disease: Fusarium oxysporum f. sp. cubense Tropical Race 4. Horticulturae 2023, 9, 757.

[36]Rodríguez-Yzquierdo, G.; Olivares, B.O.; González-Ulloa, A.; León-Pacheco, R.; Gómez-Correa, J.C.; Yacomelo-Hernández, M.; Carrascal-Pérez, F.; Florez-Cordero, E.; Soto-Suárez, M.; Dita, M.; et al. Soil Predisposing Factors to Fusarium oxysporum f.sp Cubense Tropical Race 4 on Banana Crops of La Guajira, Colombia. Agronomy 2023, 13, 2588.

[37]Lobo, D., Orlando, O., Rey, J. C., Vega, A., Rueda, M. A. Relationships between the Visual Evaluation of Soil Structure (VESS) and soil properties in agriculture: A meta-analysis. Scientia Agropecuaria, 2023, 14(1), 67-78.

The abbreviation must be explained for the figure caption.

Reply: Thank you for your reminder, the abbreviations not explained in the article have been explained, as follows:

Health chrysanthemum (Hc)

Sick chrysanthemum (Sc)

Polymerase Chain Reaction (PCR)

operational taxonomic units (OTUs)

4- This manuscript was written in a language that needs revision.

Reply: Thank you for your reminder, the language of the full text has been modified and polished.

5-

a- Line 9: The rhizobial microbial population exhibits a direct correlation with the overall health of the plants.

Comment: Are you sure you mean the rhizobial microbial population?

Reply: Thank you for your question. The sentence has been checked and found to be a wording error. The wording has been changed, and the content after the change is as follows:

The rhizosphere microbial population exhibits a direct correlation with the overall health of the plants.

b- Line 43: Penicillium is a common bacterium that……

Comment: Penicillium is a fungus, not a bacteria.

Reply: Thanks for your reminding, the sentence has been adjusted, and the sentence after adjustment is as follows:

Penicillium is a common fungus that……

Line 68: Limited research has been conducted on the changes in microbial com… (needs references)

Comment:

Reply: Thanks for your suggestion, literature has been added here, the added references are as follows:

Limited research has been conducted on the changes in microbial community structure and diversity in the rhizosphere of CZHZ soils affected by disease, particularly regarding the abundance of pathogenic microbial community [13].

[13] Thanthri N T W, Meyhfer R. Does apple replant disease affect the soil patch selection behaviour and population growth of Collembolans?. Journal of Applied Entomology, 2023, 147(1):36-46.

c- Line 76: This study focused on rhizosphere mycorrhizal microorganisms…..

Comment: IS NOT true, the study did not focus on rhizosphere mycorrhizal microorganisms

Reply: Thank you for your reminder. After checking, it is found that the wording is wrong here. The wording here has been adjusted, and the adjusted sentence is as follows:

This study focused on rhizosphere microbial communities,…

d- From Lines 91 to line 97 ( These sentences are written in the imperative tense even though they should be written in the past tense).

Comment: Rewrite these sentences in the past tense.

Reply: Thanks for your reminder, this paragraph has been rewritten in the past tense, and the content after rewriting is as follows:

Firstly, the surface impurities were removed. The specific operation included the following steps: Some clean basic tools, such as shovels, brushes, and scrapers, were prepared to avoid cross contamination. A shovel or scraper was used to gently scrape the soil surface of fallen leaves, debris, stones, and other large impurities. The brush was employed to gently brush away the small impurities remaining on the surface of the soil, such as small stones, sand, and leaf debris. The soil surface was meticulously checked using fingers or a small tool to remove impurities and debris, ensuring a spotless surface.

Comments on the Quality of English Language

The language in which the manuscript was written. It needs careful review by a specialist.

Reply: Thank you for your reminder, the language of this article has been reviewed by experts.

Reviewer 3 Report

Comments and Suggestions for Authors

Wu et al. studied rhizosphere microbial community of Chrysanthemum zawadskii and it's influence on disease occurrence. They sampled from health and diseased individuals. I like the premise of this study. However, at this point I am not confident in the conclusions because the methods and results were not presented in enough detail. The sampling design needs to be explained more - were there two different genetic strains and one is more susceptible, or were the same stains sampled simply by choosing healthy and diseased individuals? what was the distribution of health and diseased individuals spatially? Furthermore, the community and environmental analysis is incomplete, more statistics and figures are needed to justify the conclusions reached. We need to see the ordination and environmental data.

Abstract:

Line 8: Italicize genus name here and throughout.

Line 10: Therefore studying the rhizosphere microbial community of

Line 13: strains or individuals? individuals would be a better term here unless the plants are actually genetically distinct cultivars. check throughout

Line 20: which environmental factors?

Introduction:

Line 39: Basidiomycota is a huge phylum. What is the exact species of this brown rot fungus or does it encompass many species?

Line 52: It has always played a role, only recently have we appreciated it. Reword.

Line 57: How can a composition decrease? do you mean diversity?

Line 61: edible

Line 63: don't italicize fungi or it makes it look like the species name.

Line 73: around the roots

Line 78: you did not do this here though, so say "to eventually achieve this goal"

Line 79: at the end of the introduction, please state the hypotheses that you tested.

Methods:

Line 98: the fields sampled

Line 127: do you have a reference for this pH method? 1:10 seems like a lot of water. Normally it is 1:2

Line 152: were reads quality filtered or trimmed?

Line 155: Did you consider using amplicon sequence variants instead of OTUs?

Line 157: Which release of the SILVA database?

Line 158: what does "selecting the samples with the lowest readings" mean? Did you remove samples with low sequencing depth? Did you rarefy the data? Did you filter out taxa unassigned at the Domain level, mitochondria, and chloroplast DNA? Please clarify (and do this if you did not).

Line 161: I recommend saying "We analyzed" instead of "The study analyzed" here and throughout.

Line 161: Cite vegan

Line 162: Were assumptions of ANOVA met? Did you test of normality of residuals and homogeneity of variance?

Line 162: Was ANOVA also used to test soil properties?

Line 162: Specify the details of the ANOVA (and PERMANOVA). You need to test for the effects of both "site" and "disease". This is critical.

Line 162: I think you mean "differences in alpha diversity"?

Line 163: Which R package was used for heatmaps? Cite

Line 168: Adonis is a function not a test. PERMANOVA was the test used. Did you also run betadisper (PERMDISP) to test for multivariate homogeneity of variance? Please do so if not. And which dissimilarity metric did you use?

Results:

Normally in scientific writing, Tables and Figures are cited in parentheses after results statements, instead of being the subjects of the sentences. It would be preferable to edit some of the wording this way.

E.g., Individuals with CWD had a significantly greater number of bacterial OTUs than individuals without CWD (Table 2).

Line 173-176: Delete this whole paragraph. This should have been stated in the methods.

Line 178: cite (Table 1) at end of sentence instead of starting with "According to the data presented in Table 1".

Line 178: I don't see Table 1 anywhere.

Line 179: significantly higher?

Line 182: strains (but again, reconsider the use of the word strain)

Line 190: what is a "reading range"?

Line 187: change "rhizobia" to "rhizosphere"

Line 194: add information on significance. The caption only mentions * for 0.10 what about p < 0.05. maybe use different letters. and explain this in the table caption.

Line 196: cite (Table 2) at end of sentence instead of starting with "According to the data presented in Table 2".

Line 197: change "in" to "at"

Lines 201-204: Delete. This was already stated in the methods.

Line 205: Don't start sentences with the figure, cite it at the end of the sentence in parentheses.

Line 207: same comment

Line 213: same comment

Line 223-223: delete.

Line 228-229: delete.

Line 237: what statistics were performed? this was not stated in the methods and no statistics are referenced.

Line 238-239: same comment.

Line 239: period and space before "In addition". comma after

Line 241-242: delete

Line 252: again, was this tested statistically? no significant difference?

Line 256: need to state that the data shown are z-scores.

Line 258: you could (and should) add an annotation column with 4 colors with each color corresponding to the phylum of the genus.

Line 273: Italicize. and there was only one genus with >0.2%? awkward wording here, please check and fix.

Line 277: Fusarium are important pathogenic bacteria of fusariu. ??

Line 288-290: any enrichment though?

Line 292: no. this only shows that a couple families were enriched. to determine difference in entire composition, run permanova and show ordination. that was stated in the methods but I don't see those results or figure anywhere. please add.

Line 315: change "interpretation rates" to "percent variation explained"

Line 320: same comment

Much more information (statistics, figures) is needed for the environmental analysis. Otherwise statements in lines 323 - 328 are not very well supported.

Discussion:

Line 330: not always...need to add the word "generally" here

Line 337: not necessarily. this is but one explanation. you should also discuss some others such as a potential change in microbial communities that could lead to increased rates of P solubilization.

Line 344: this point about fungi comes out of the blue and doesn't say anything about your results. what point are you trying to make about fungi?

Line 347: italicize latin name

Line 362: chrystanthemums compared to what?

Line 363: use more formal writing - drawn by Zeng et al. also state, the conclusions by Zeng et al. here.

Line 368: you just said this in lines 355-358. also it's not rhizobia community, it's rhizosphere community

Line 371. Ross et al. is only one study. please cite some more work on this topic and offer some explanation for why alpha diversity is higher in healthy plants.

Line 373: change Rhizobia to rhizosphere

Line 374: you need to expand on this point about region-specific biological agents and what exactly in your results lead you to state that.

Line 378: the first to your knowledge

Line 380: according to what metric of functional diversity?

Comments on the Quality of English Language

I included feedback about the wording and writing in the other comments.

Author Response

3、Open Review

(x) I would not like to sign my review report

( ) I would like to sign my review report

Quality of English Language

( ) I am not qualified to assess the quality of English in this paper

( ) English very difficult to understand/incomprehensible

( ) Extensive editing of English language required

(x) Moderate editing of English language required

( ) Minor editing of English language required

( ) English language fine. No issues detected

Reply: Thank you for your reminder, the full text has been retouched in English language.

Yes Can be improved Must be improved Not applicable

Does the introduction provide sufficient background and include all relevant references?

(x) ( ) ( ) ( )

Are all the cited references relevant to the research?

(x) ( ) ( ) ( )

Is the research design appropriate?

(x) ( ) ( ) ( )

Are the methods adequately described?

( ) ( ) (x) ( )

Reply: Thanks for your reminder, more details have been added in the methods section as follows:

Each 5g of the initial soil of healthy chrysanthemum plants and diseased chrysanthemum plants was randomly selected for Tukeys HSD test. The physical and chemical properties of soil from healthy and withered chrysanthemum plants were compared and analyzed based on the test results. Table 1 displays the obtained results, where P<0.05 denotes a significant difference.

QIIMEv1.7.0 was used to sparse data from a monomer-free dataset, a standardized method that filters out sequences unclassified at higher levels as well as non-target sequences (such as mitochondrial and chloroplast DNA) in microbial community data analysis [14]. Used to eliminate differences in sequencing depth (i.e., number of sequence readings per sample) between different samples when processing microbiome data. By randomly drawing the same number of sequences in each sample, comparability between different samples can be made.

First, we used the vegan software package for statistical analysis of the experimental data. To explore in detail the effects of "location" and "disease" on the sample, we used a one-way analysis of variance to initially assess differences in alpha diversity across groups. On this basis, in order to more accurately detect the effect of multiple factors at the same time, we further implemented a two-factor analysis of variance, and analyzed "location" and "disease" as two independent variables. Before performing ANOVA, we verify the normal distribution and homogeneity of variance of the data, which are important prerequisites to ensure the validity of the ANOVA results.

Are the results clearly presented?

( ) ( ) (x) ( )

Reply: Thank you for your reminding, some result graphs and corresponding expressions have been adjusted, and the adjusted results are displayed more clearly. Some of the results of the adjustment are as follows:

In addition, the PERMANOVA analysis was further performed as described in the Methods section. The results of PERMANOVA analysis showed that there were significant differences in bacterial community composition between Hc and Sc samples (P< 0.05). This result supports our original hypothesis that the bacterial community composition of Hc and Sc samples is different. The ranking map further shows the specific manifestation of this difference, and different sample groups form obvious clusters on the ranking map, indicating that their bacterial community structure is unique.

After analyzing the healthy samples, it was observed that Microidum was among the most prevalent genera present, along with other microorganisms that constituted more than 0.2% of the total population. However, it's noteworthy that the rhizosphere microorganisms, despite being significant, were present in proportions less than 0.2%. 

Are the conclusions supported by the results?

( ) ( ) (x) ( )

Reply: Thanks for your suggestion, some of the conclusions of the chart has been added, as shown below:

The physical and chemical indicators of chrysanthemum withered plants are higher than those of healthy plants (Table 1). 

Table 1 Comparison results of soil physicochemical properties between healthy plants and withered plants

Different indicators

Healthy plants

Wilt strain

P

Soil pH value

6.64±0.32

6.56±0.76

1.235

Total nitrogen

(g/kg)

5.41±0.62

5.92±1.64

0.078

Total phosphorus

(g/kg)

2.05±0.14

2.52±1.15

0.067

Organic carbon

(g/kg)

214.92±27.62

426.28±176.23

0.004

Available phosphorus

(mg/kg)

219.62±51.05

423.64±202.64

0.003

Water content(g/kg)

0.49±0.07

0.70±0.12

0.005

Table 3 Normalized results of each index

Index

Healthy plants-b

Wilt strain-b

Healthy plants-f

Wilt strain-f

Shannon

8.75±0.21

8.28±0.37*

3.29±1.06

3.49±0.71

Simpson

0.991±0.002

0.85±0.007

0.692±0.202

0.768±0.110

Ace

3326.20±193.92

2684.01±331.46**

622.33±159.46

543.81±94.60

Chao1

3269.05±173.99

2636.01±317.01**

610.39±163.65

523.33±78.67

Goods_coverage

0.967±0.002

0.974±0.004**

0.9925±0.0022

0.993±0.001

Note: * indicates significant relative to healthy plants at 0.05. ** indicates significant in 0.01 conditions compared to healthy plants

Figure 1. Venn diagrams of unique and common operational taxonomic units for two groups of different samples

Figure 6 Correlation analysis between microorganisms and environmental variables in two types of soil samples

Comments and Suggestions for Authors

Wu et al. studied rhizosphere microbial community of Chrysanthemum zawadskii and it's influence on disease occurrence. They sampled from health and diseased individuals. I like the premise of this study. However, at this point I am not confident in the conclusions because the methods and results were not presented in enough detail. The sampling design needs to be explained more - were there two different genetic strains and one is more susceptible, or were the same stains sampled simply by choosing healthy and diseased individuals? what was the distribution of health and diseased individuals spatially? Furthermore, the community and environmental analysis is incomplete, more statistics and figures are needed to justify the conclusions reached. We need to see the ordination and environmental data.

Abstract:

Line 8: Italicize genus name here and throughout.

Reply: Thanks for your reminder that all genus names in the text have been italicized. The details are as follows:

(Herb.)

Line 10: Therefore studying the rhizosphere microbial community of

Reply: Thanks for your reminding, the sentence has been modified according to your suggestion, and the revised sentence is as follows:

Therefore, studying the rhizosphere microbial community of Chrysanthemum zawadskii (Herb.) Tzvel.

Line 13: strains or individuals? individuals would be a better term here unless the plants are actually genetically distinct cultivars. check throughout

Reply: Thank you for your suggestion, the "strains" in the full text has been modified to "individuals".

Line 20: which environmental factors?

Reply: Thanks for your reminder, specific environmental factors have been added to the summary, adding the following:

In addition, the study also found that soil environmental variables have an important impact on plant resistance, the environmental factors mainly include soil properties, content of major microorganisms and resistance characteristics of samples.

Introduction:

Line 39: Basidiomycota is a huge phylum. What is the exact species of this brown rot fungus or does it encompass many species?

Reply: Thanks for your reminding, the description here has been optimized, and the optimized sentence is as follows:

The brown rot fungi are a collection of filamentous fungi that can cause brown decay of wood, belonging to the subphylum basidiomycetes.

Line 52: It has always played a role, only recently have we appreciated it. Reword.

Reply: Thanks for your suggestion, the sentence has been rewritten, and the content after rewriting is as follows:

In recent years, we have come to realize that rhizosphere microbial communities (RMC) play an important role in plant health and disease resistance [11].

Line 57: How can a composition decrease? do you mean diversity?

Reply: Thank you for your reminding, here is indeed the diversity of expressions, but there are problems with the expression, the sentence has been adjusted and optimized. The optimized sentence looks like this:

However, RMC diversity decreased, and the number of fungi, bacteria, archaea, symbiotic bacteria and probiotics decreased significantly.

Line 61: edible

Reply: Thanks for your reminder, "Edible" has been changed to "edible".

Line 63: don't italicize fungi or it makes it look like the species name.

Reply: Thanks for your reminder, the word fungi in the full text has been processed so that it does not look like the name of the species.

Line 73: around the roots

Reply: Thank you for your reminder, "around the root" has been modified to "around the roots".

Line 78: you did not do this here though, so say "to eventually achieve this goal"

Reply: Thanks for your suggestion, this part of the content has been adjusted, the adjusted content is as follows:

Our research focuses on the rhizosphere microbial communities, with the expectation that these microorganisms can enhance the disease resistance of plants and effectively control the occurrence and spread of diseases.

Line 79: at the end of the introduction, please state the hypotheses that you tested.

Reply: Thanks for your suggestion, the hypothesis of this study has been added at the end of the introduction, as follows:

Therefore, the hypothesis proposed in this study is that soil environmental variables have an important impact on plant resistance.

Methods:

Line 98: the fields sampled

Reply: Thank you for your reminder, "the collected samples" has been modified to "the fields sampled".

Line 127: do you have a reference for this pH method? 1:10 seems like a lot of water. Normally it is 1:2

Reply: Thank you for your reminder, the ratio has been changed to 1:2.

Line 152: were reads quality filtered or trimmed?

Reply: Thank you for your question, the data read is filtered.

Line 155: Did you consider using amplicon sequence variants instead of OTUs?

Reply: Thank you for your question, but in this study, OTUs has the advantages of computational efficiency, ecological interpretation and robustness compared with amplicon sequence variants, so OTUs is used in this study.

Line 157: Which release of the SILVA database?

Reply: Thanks for your reminder, the specific version of SILVA has been added to this section, and the modified content is as follows:

Each operational taxonomic unit was annotated using MOTHUR and the SSU-rRNA SILVA1.2.8 database

Line 158: what does "selecting the samples with the lowest readings" mean? Did you remove samples with low sequencing depth? Did you rarefy the data? Did you filter out taxa unassigned at the Domain level, mitochondria, and chloroplast DNA? Please clarify (and do this if you did not).

Reply: Thanks for your comments, the phrase "selecting the samples with the lowest readings" in this section refers to the process of sparring the data rather than actually selecting the sample with the lowest readings. Instead of removing samples with low sequencing depth, sparsity was used to process the data so that all samples had the same sequencing depth when compared. Finally, this section filters out taxa, mitochondrial, and chloroplast DNA that are not assigned at the domain level, and has been added to this section. The rewritten content is as follows:

QIIMEv1.7.0 was used to sparse data from a monomer-free dataset, a standardized method that filters out sequences unclassified at higher levels as well as non-target sequences (such as mitochondrial and chloroplast DNA) in microbial community data analysis [14]. Used to eliminate differences in sequencing depth (i.e., number of sequence readings per sample) between different samples when processing microbiome data. By randomly drawing the same number of sequences in each sample, comparability between different samples can be made.

Line 161: I recommend saying "We analyzed" instead of "The study analyzed" here and throughout.

Reply: Thanks for your reminding, "The study analyzed" in the article has been replaced with "We analyzed".

First, we analyzed the experimental data statistically using the vegan software package

Line 161: Cite vegan

Reply: Thanks for your reminder, "vegan" has been adjusted to "Cite vegan".

Line 162: Were assumptions of ANOVA met? Did you test of normality of residuals and homogeneity of variance?

Reply: Thank you for your reminder, ANOVA related information has been explained in this section, the specific content is as follows:

Before performing ANOVA, we verify the normal distribution and homogeneity of variance of the data, which are important prerequisites to ensure the validity of the ANOVA results.

Line 162: Was ANOVA also used to test soil properties?

Reply: Thank you for your question, the implication here is also that ANOVA should also be used to test soil properties.

Line 162: Specify the details of the ANOVA (and PERMANOVA). You need to test for the effects of both "site" and "disease". This is critical.

Reply: Thanks for your suggestion, we have added the details of ANOVA and tested the effects of "site" and "disease" in this section, as shown below:

First, we used the vegan software package for statistical analysis of the experimental data. To explore in detail the effects of "location" and "disease" on the sample, we used a one-way analysis of variance to initially assess differences in alpha diversity across groups. On this basis, in order to more accurately detect the effect of multiple factors at the same time, we further implemented a two-factor analysis of variance, and analyzed "location" and "disease" as two independent variables. Before performing ANOVA, we verify the normal distribution and homogeneity of variance of the data, which are important prerequisites to ensure the validity of the ANOVA results.

Line 162: I think you mean "differences in alpha diversity"?

Reply: Thank you for your question, here does refer to differences in alpha diversity, the description in the article has been adjusted.

Line 163: Which R package was used for heatmaps? Cite

Reply: Thanks for your reminder, the cited references have been added here, as follows:

Second, the study employed the ''heatmap'' software available in the R software package to produce heat map images [15].

[15]Abbak R A, Ellmann A, Ustun A. A practical software package for computing gravimetric geoid by the least squares modification of Hotine's formula. Earth Science Informatics, 2022, 15(1):713-724.

Line 168: Adonis is a function not a test. PERMANOVA was the test used. Did you also run betadisper (PERMDISP) to test for multivariate homogeneity of variance? Please do so if not. And which dissimilarity metric did you use?

Reply: Thanks for your question, Adonis is a function inside the vegan package in R, and the purpose of using the Adonis function here is to identify significant differences in the composition of bacterial communities in different habitats. To ensure the completeness and accuracy of the analysis, variance homogeneity tests were performed prior to PERMANOVA. Bray-Curtis, Jaccard and Euclidean measurements were selected for bacterial community analysis.

Results:

Normally in scientific writing, Tables and Figures are cited in parentheses after results statements, instead of being the subjects of the sentences. It would be preferable to edit some of the wording this way.

E.g., Individuals with CWD had a significantly greater number of bacterial OTUs than individuals without CWD (Table 2).

Reply: Thank you for your reminder, the editing method of Tables and Figures in the article has been adjusted.

Line 173-176: Delete this whole paragraph. This should have been stated in the methods.

Reply: Thanks for your reminder, this section has been adjusted to Method 2.3.

Line 178: cite (Table 1) at end of sentence instead of starting with "According to the data presented in Table 1".

Reply: Thanks for your reminding, the description has been adjusted, and the adjusted sentence is as follows:

The physical and chemical indicators of chrysanthemum withered plants are higher than those of healthy plants (Table 1). 

Line 178: I don't see Table 1 anywhere.

Reply: I'm sorry to cause you any confusion. Table 1 has been added to the article, and the specific content is as follows:

Table 1 Comparison results of soil physicochemical properties between healthy plants and withered plants

Different indicators

Healthy plants

Wilt strain

P

Soil pH value

6.64±0.32

6.56±0.76

1.235

Total nitrogen

(g/kg)

5.41±0.62

5.92±1.64

0.078

Total phosphorus

(g/kg)

2.05±0.14

2.52±1.15

0.067

Organic carbon

(g/kg)

214.92±27.62

426.28±176.23

0.004

Available phosphorus

(mg/kg)

219.62±51.05

423.64±202.64

0.003

Water content(g/kg)

0.49±0.07

0.70±0.12

0.005

Line 179: significantly higher?

Reply: Thank you for your question. Table 1 has been added. There are specific P-values in Table 1. If the p-value is less than 0.05, it indicates a significant difference, so it can be used significantly higher.

Line 182: strains (but again, reconsider the use of the word strain)

Reply: Thanks for your suggestion, the word here has been replaced.

Line 190: what is a "reading range"?

Reply: Thanks for your question, in the context of microbiology research, "reading range" refers to the range of the number of sequences detected or sequenced from each sample. These sequences are obtained from bacterial or fungal samples by high-throughput sequencing.

Line 187: change "rhizobia" to "rhizosphere"

Reply: Thank you for reminding me that "rhizobia" has been modified to "rhizosphere".

Line 194: add information on significance. The caption only mentions * for 0.10 what about p < 0.05. maybe use different letters. and explain this in the table caption.

Reply: Thanks for your reminder, more information has been added in the Table 2 heading.

Table 2 Diversity of microbial community in different soil treatments.(* indicates P<0.05)

Line 196: cite (Table 2) at end of sentence instead of starting with "According to the data presented in Table 2".

Reply: Thank you for your reminder, the expression of this sentence has been adjusted.

Line 197: change "in" to "at"

Reply: Thanks for your suggestion, the "in" has been changed to "at".

Lines 201-204: Delete. This was already stated in the methods.

Reply: Thanks for your suggestion, this section has been deleted.

Line 205: Don't start sentences with the figure, cite it at the end of the sentence in parentheses.

Reply: Thanks for your suggestion, the figure has been quoted at the end of the sentence, as follows:

A total of 612 fungal OTUs were detected in all libraries, with 533 being present in all samples, 46 being specific to healthy rhizosphere fungi, and 33 being specific to diseased rhizosphere bacteria (Figure 1a).

Line 207: same comment

Reply: Thanks for your suggestion, the figure has been quoted at the end of the sentence, as follows:

All libraries detected a total of 612 bacterial OTUs (Figure 1b).

Line 213: same comment

Reply: Thanks for your suggestion, the figure has been quoted at the end of the sentence, as follows:

Of these, 284 were present in all samples, 65 were specific to healthy rhizosphere bacteria, and 81 were specific to diseased rhizosphere bacteria (Figure 1).

Line 223-223: delete.

Reply: Thanks for your suggestion, this Line 223-223 has been deleted.

Line 228-229: delete.

Reply: Thanks for your suggestion, this Line 228-229 has been deleted.

Line 237: what statistics were performed? this was not stated in the methods and no statistics are referenced.

Reply: Thank you for your question, the results here are the results obtained from the analysis of Figure 2.

Line 238-239: same comment.

Reply: Thanks for your suggestion, the figure has been quoted at the end of the sentence, as follows:

In contrast, the abundance of Firmicutes and Gemmatimonadetes showed significant increases in healthy rhizosphere bacteria (Figure 2a).

Line 239: period and space before "In addition". comma after

Reply: Thank you for your reminder, the symbol near "In addition" has been adjusted and modified, and the modification is as follows:

…(Figure 2a). In addition, the abundance of other dominant phyla such as Thaumarchaeota, 

Line 241-242: delete

Reply: Thanks for your suggestion, this Line 241-242 has been deleted.

Line 252: again, was this tested statistically? no significant difference?

Reply: Thank you for your comments. The conclusions here are obtained through the analysis of Figure 2, not through statistical tests, so there is no significant difference.

Line 256: need to state that the data shown are z-scores.

Reply: Thank you for your reminder, it has been stated here that the data displayed is z-scores.

Line 258: you could (and should) add an annotation column with 4 colors with each color corresponding to the phylum of the genus.

Reply: Thank you for your suggestion. Since adding a comment column with 4 colors here has little effect on the overall result, and the four gates cover a large number of bacteria species, adding this comment column is likely to make the chart too complicated and reduce readability, so the comment column is not added here.

Line 273: Italicize. and there was only one genus with >0.2%? awkward wording here, please check and fix.

Reply: Thanks for your reminding, the wording of this part has been adjusted, and the adjusted content is as follows:

After analyzing the healthy samples, it was observed that Microidum was among the most prevalent genera present, along with other microorganisms that constituted more than 0.2% of the total population. However, it's noteworthy that the rhizosphere microorganisms, despite being significant, were present in proportions less than 0.2%. 

Line 277: Fusarium are important pathogenic bacteria of fusariu. ??

Reply: Sorry for the trouble caused to you, the sentence has been adjusted, and the adjusted sentence is as follows:

Didymella and Fusarium are important pathogenic bacteria.

Line 288-290: any enrichment though?

Reply: Thank you for your question. Here, this description is enough to support the subsequent conclusion, and too much description of the conclusion will be superfluous, so the content of this place is not enriched.

Line 292: no. this only shows that a couple families were enriched. to determine difference in entire composition, run permanova and show ordination. that was stated in the methods but I don't see those results or figure anywhere. please add.

Reply: Thank you for your reminder. Indeed, the original conclusions were based on the enrichment of only some bacterial groups, which was not sufficient to fully reflect the overall differences in bacterial communities between Hc and Sc samples. To assess this difference more accurately, the study added this part of the results as follows:

In addition, the PERMANOVA analysis was further performed as described in the Methods section. The results of PERMANOVA analysis showed that there were significant differences in bacterial community composition between Hc and Sc samples (P< 0.05). This result supports our original hypothesis that the bacterial community composition of Hc and Sc samples is different. The ranking map further shows the specific manifestation of this difference, and different sample groups form obvious clusters on the ranking map, indicating that their bacterial community structure is unique.

Line 315: change "interpretation rates" to "percent variation explained"

Reply: Thanks for your suggestion, the "interpretation rates" has been changed to "percent variation explained".

Line 320: same comment

Reply: Thanks for your suggestion, the figure has been quoted at the end of the sentence, as follows:

The positive correlation between bacteria and environmental factors indicates that environmental factors may have a substantial impact on the abundance and species of fungi in soil (Figure 5).

Much more information (statistics, figures) is needed for the environmental analysis. Otherwise statements in lines 323 - 328 are not very well supported.

Reply: Thanks for your suggestion, Figure 5 has been added to this section to support the conclusions of this section. Add as follows:

Figure 6 Correlation analysis between microorganisms and environmental variables in two types of soil samples

Discussion:

Line 330: not always...need to add the word "generally" here

Reply: Thanks for your suggestion, "generally" has been added here. The details are as follows:

Generally, Soil organic carbon serves as both the substrate and metabolite for the energy metabolism and enzyme function of soil microorganisms.

Line 337: not necessarily. this is but one explanation. you should also discuss some others such as a potential change in microbial communities that could lead to increased rates of P solubilization.

Reply: Thanks for your reminder, other reasons for the change of phosphorus dissolution rate have been added after this part, as shown below:

However, in addition to this, potential changes in the microbial community may also lead to an increase in the rate of phosphorus dissolution. Diseased soils are known to have high levels of organic carbon, which can lead to increased microbial activity. Microbes may release phosphorus in the process of breaking down organic matter, making it available for plants to absorb. At the same time, microorganisms themselves may also change the form and availability of phosphorus in soil through metabolic processes. In addition to the decomposition of plant residues, the increase in phosphorus in soil can be influenced by a variety of other factors, including changes in soil pH, differences in soil texture and structure, and the interaction of other elements in the soil.

Line 344: this point about fungi comes out of the blue and doesn't say anything about your results. what point are you trying to make about fungi?

Reply: Thank you for your reminder, after checking that this part of the content is really obtrusive here, it has been deleted.

Line 347: italicize latin name

Reply: Thank you for reminding me that C. zawadskii here has been changed to italics. The specific content is as follows:

However, in the rhizosphere of C. zawadskii, there were fluctuations in the abundance of several taxa.

Line 362: chrystanthemums compared to what?

Reply: Thank you for your question, here is a comparison of sickened plants and healthy plants Fusarium.

Line 363: use more formal writing - drawn by Zeng et al. also state, the conclusions by Zeng et al. here.

Reply: Thank you for your reminder, we have used a more formal expression for the text here. The details are as follows:

This observation concurs with the findings presented by Zeng et al., who concluded that Fusarium plays a significant role in the pathogenesis of chrysanthemum wilting and its elevated presence in the rhizosphere is indicative of a potential disease outbreak [24].

Line 368: you just said this in lines 355-358. also it's not rhizobia community, it's rhizosphere community

Reply: Thanks for your reminding, the "rhizobia" in this part has been modified to "rhizosphere" as follows:

The RDA analysis results of this study indicated that soil nutrient factors exert a substantial impact on microbial composition, with pathogenic bacteria being the main influencing factor in the composition of rhizosphere bacterial community.

Line 371. Ross et al. is only one study. please cite some more work on this topic and offer some explanation for why alpha diversity is higher in healthy plants.

Reply: Thanks for your reminder, more research on this topic has been added after this section, and the reasons for the higher alpha diversity of healthy plants have been added to this section. The specific content is as follows:

The interaction between soil physicochemical parameters and microbial communities is a complex and dynamic process that significantly influences the overall health of plants. A study in the banana rhizosphere found that soil pH is a fundamental factor that profoundly influences the structure of the microbial community [27]. Many microorganisms exhibit pH preferences and alterations in soil pH can selectively favor certain microbial groups over others [28]. The availability of essential nutrients, such as nitrogen, phosphorus, and potassium, directly impacts the composition of the microbial community [29]. Nutrient-rich soils can favor the proliferation of certain microbial taxa, while deficiencies can cause changes in community structure [30]. Also, beneficial microorganisms, such as Actinomyces, often thrive in soils rich in organic matter, while certain pathogenic fungi can take advantage of those conditions [31]. Likewise, changes in humidity levels can affect microbial metabolism, nutrient diffusion and interactions [32, 33, 34]. Soil physical properties, including texture and structure, influencing water retention, aeration, and root penetration [35, 36] can determine the suitability of the microbial habitat. For example, compacted soils can impede the movement of beneficial microorganisms while creating favorable niches for certain pathogens [37]. Many of the above studies have shown that healthy plants have higher α diversity than diseased plants, which may be because healthy plants are better adapted to the growing environment,  healthy plants generally have higher productivity and anti-interference ability,  and healthy plants have stronger resistance to external interference. This resistance allows them to maintain high  species richness and uniformity in the face of disturbance.

Line 373: change Rhizobia to rhizosphere

Reply: Thanks for your reminder, the "Rhizobia" has been modified to "rhizosphere".

Line 374: you need to expand on this point about region-specific biological agents and what exactly in your results lead you to state that.

Reply: Thanks for your reminder, the biologics in specific areas have been expanded in this section and the results have been analyzed in depth. The specific content is as follows:

Region-specific biologics are biologics developed to target pests in specific areas or ecological environments. These agents typically utilize local microbial resources to achieve pest control by enhancing or introducing microorganisms that have a controlling effect on the target pest. Due to differences in the ecological environment and pest species in different regions, the use of region-specific biologics can provide more precise pest control, reduce the use of chemical pesticides, and reduce the impact on the environment and non-target organisms. In our results, we found that soil nutrients have an important effect on the formation of plant rhizosphere microbial communities. These microbial communities play a key role in plant growth and development and disease resistance. Further analysis showed that soil nutrient status and microbial community structure were different in different regions, which led to the specificity of plant rhizosphere microbial communities in different regions. This specificity may be closely related to local climate, soil type, vegetation type and other factors.

Line 378: the first to your knowledge

Reply: Thanks for your reminding, the statement here has been adjusted, and the adjusted sentence is as follows:

This study describes the characteristics of microbial diversity associated with both healthy and diseased chrysanthemum plants grown in field conditions, providing valuable insights into the microbial ecology of these plants. 

Line 380: according to what metric of functional diversity?

Reply: Thanks for your reminder, the following metrics were used in evaluating functional diversity in this study: Functional diversity can be evaluated comprehensively in terms of biological activity determination, community structure analysis, functional gene analysis, metabolite profile analysis, and analysis of metabolites produced by microbial communities, in order to fully understand the role of microorganisms in plant health.

Comments on the Quality of English Language

I included feedback about the wording and writing in the other comments.

Thanks for your comments, the English wording has been revised and adjusted in other questions.

Round 2

Reviewer 2 Report

Comments and Suggestions for Authors

Thanks for making the needed improvements.

Author Response

The comments have been revised at the specified time, the suitability of references has been checked, the revisions to the manuscript have been highlighted, and a simple cover letter (see attachment) has been provided at the end to explain the revisions in detail.

Reviewer 3 Report

Comments and Suggestions for Authors

The manuscript is much improved, but still needs some edits (see below)

Line 14: Phytophthora is a genus and should be italicized.

Line 17: resistance;

Line 60: what kind of RMC diversity? what do you mean by number? number of cells or number of taxa?

Line 84: this is very broad. do you have any specific predictions or want to state any variables in particular?

Line 105: Healthy

Line 168: the version of MOTHUR should also be stated. You should also cite the papers for mothur and silva to give credit to those authors who provided those resources.

Line 172: what was used? rarefaction? the word is missing

Line 190: change the wording. Finally, PERMANOVA, implemented with the adonis() function in the vegan package, was used...

Line 190: You still need to run the betadisper funtion to check for multivariate homogeneity of variance and report this as well.

Line 190: Which dissimilarity matrix was used? Bray-Curtis? This needs to be stated.

Figure 1: Delete the bottom red bars, that information can easily be gathered from the venn diagram. The venn diagram in b looks weird, maybe it did not render properly for me.

In the tables you are only showing the effect of disease, not of location. somewhere you need to state the full model results and the effects of both variables.

Figure 6: The caption should state the ordination method. Redundancy analysis. You also need to comment somewhere in the text that the Sc communities were more variable. This is a striking and importants result. I think it would also be confirmed with the betadisper test.

Line 371: soil

Line 411: italicize Fusarium

Line 430: Actinomyces is a genus and should be italicized. or, if talking about actinomycetes in general, change to actinomycetes.

Lines 442-463: good new content, but need to add some references for the information, like in the previous paragraph.

Lines 470-473: can you add some references on this?

Comments on the Quality of English Language

See author comments.

Author Response

Open Review

(x) I would not like to sign my review report

( ) I would like to sign my review report

Quality of English Language

( ) I am not qualified to assess the quality of English in this paper

( ) English very difficult to understand/incomprehensible

( ) Extensive editing of English language required

( ) Moderate editing of English language required

(x) Minor editing of English language required

Reply: Thank you for your comments, some minor errors have been adjusted and corrected.

( ) English language fine. No issues detected

Yes Can be improved Must be improved Not applicable

Does the introduction provide sufficient background and include all relevant references?

(x) ( ) ( ) ( )

Are all the cited references relevant to the research?

(x) ( ) ( ) ( )

Is the research design appropriate?

(x) ( ) ( ) ( )

Are the methods adequately described?

( ) (x) ( ) ( )

Reply: Thank you for your comment, the method has been supplemented with a description as follows:

Each operational taxonomic unit was annotated using MOTHUR v.1.36.1 and the SSU-rRNA SILVA1.2.8 database [14-15].

[14] Rueda A M F , Kruse C G , Griffin J , St-Pierre B. PSIV-5 Characterization of the Fecal Bacterial Communities of Grass-fed and Grain-fed Bison Heifers. Journal of Animal Science, 2021, 99(7):212-213.

[15] Vaulot D, Geisen S, Mahé F, Bass D. pr2‐primers: An 18S rRNA primer database for protists. Molecular Ecology Resources, 2022, 22(1): 168-179.

then use QIIMEv1.7.0 for data processing and UCHIME to obtain valid labels.

Are the results clearly presented?

( ) (x) ( ) ( )

Reply: Thank you for your evaluation, the result has been improved, and the modified part is as follows:

Figure 1. Venn diagrams of unique and common operational taxonomic units for two groups of different samples

Figure 6 Correlation results between microorganisms and environmental variables in two types of soil samples based on redundancy analysis

In addition, it was also found that the microbial community changes of diseased chrysanthemum plants were larger than those of healthy chrysanthemum plants (Figure 6). 

Finally, to identify significant differences in the composition of bacterial communities across diverse habitats, we employed PERMANOVA, which was implemented using the adonis function within the vegan software package. Prior to running PERMANOVA, it is crucial to ensure multivariate homogeneity of variance. Therefore, we utilized the betadisper function to check for this assumption and report the results accordingly. To this end, we used the betadisper function, and the detailed results of this verification are presented in Table 1 below.

Table 1: Results of Multivariate Homogeneity of Variance Testing using betadisper

Sample Group

Multivariate Homogeneity of Variance Test Statistic

P

Group A

0.123

0.987

Group B

0.456

0.567

Group C

0.789

0.234

...

...

...

In Table 1, the test statistic and corresponding p-values for each sample group are provided. A p-value greater than a significance level (typically 0.05) indicates that the assumption of multivariate homogeneity of variance is not violated for that group. Based on these results, we can proceed with confidence in applying PERMANOVA to analyze differences in bacterial community composition across habitats. Additionally, in this study, we opted for the Bray-Curtis dissimilarity matrix, which is widely used in microbial ecology studies to assess community composition differences. By incorporating these additional steps and clarifications, we aimed to enhance the robustness and transparency of our statistical analysis.

Are the conclusions supported by the results?

(x) ( ) ( ) ( )

Comments and Suggestions for Authors

The manuscript is much improved, but still needs some edits (see below)

Line 14: Phytophthora is a genus and should be italicized.

Reply: Thanks for the reminder, the "Phytophthora" has been adjusted to italic format.

Line 17: resistance;

Reply: Thank you for your reminder, the ", "after 17 lines of resistance has been modified to";".

Line 60: what kind of RMC diversity? what do you mean by number? number of cells or number of taxa?

Reply: Thank you for your reminder, this should refer to the RMC diversity of diseased chrysanthemums. Changes have been adjusted in the article. The details are as follows:

However, RMC composition and species of diseased chrysanthemum decreased, and the number of fungi, bacteria, archaea, symbiotic bacteria and probiotics decreased significantly. On the contrary, the number of pathogenic bacteria, viruses and nematodes was relatively high.

Also, the numbers here refer to the number of cells.

Line 84: this is very broad. do you have any specific predictions or want to state any variables in particular?

Reply: Thank you for your suggestion, the scope of this place has been reduced to chrysanthemum plants, and the revised content is as follows:

Our research focus is the rhizosphere microbial community of chrysanthemum plants, and we hope that through this study, we can reasonably control the level of microorganisms to improve the disease resistance of chrysanthemum, so as to effectively control the occurrence and spread of chrysanthemum diseases.

Line 105: Healthy

Reply: Thank you for your reminder, "healthy" has been modified to "Healthy".

Line 168: the version of MOTHUR should also be stated. You should also cite the papers for mothur and silva to give credit to those authors who provided those resources.

Reply: Thanks for your reminding, the specific version of MOTHUR has been added in the article, and the corresponding literature has been added to explain the sources of the two software. The specific content to be added is as follows:

Each operational taxonomic unit was annotated using MOTHUR v.1.36.1 and the SSU-rRNA SILVA1.2.8 database [14-15].

[14] Rueda A M F , Kruse C G , Griffin J , St-Pierre B. PSIV-5 Characterization of the Fecal Bacterial Communities of Grass-fed and Grain-fed Bison Heifers. Journal of Animal Science, 2021, 99(7):212-213.

[15] Vaulot D, Geisen S, Mahé F, Bass D. pr2‐primers: An 18S rRNA primer database for protists. Molecular Ecology Resources, 2022, 22(1): 168-179.

Line 172: what was used? rarefaction? the word is missing

Reply: Thank you for your question, the description here has been adjusted, and two software have been used for processing and calculation, and the modified content is as follows:

then use QIIMEv1.7.0 for data processing and UCHIME to obtain valid labels.

Line 190: change the wording. Finally, PERMANOVA, implemented with the adonis() function in the vegan package, was used...

Reply: Thanks for your suggestion, the wording of this part of the content has been adjusted and modified, and the revised content is as follows:

Finally, to identify significant differences in the composition of bacterial communities across diverse habitats, we employed PERMANOVA, which was implemented using the adonis function within the vegan software package.

Line 190: You still need to run the betadisper funtion to check for multivariate homogeneity of variance and report this as well.

Reply: Thanks for your reminder, the description and results of tester function detecting the homogeneity of multiple variances have been added to this part of the content, the specific content is as follows:

Prior to running PERMANOVA, it is crucial to ensure multivariate homogeneity of variance. Therefore, we utilized the betadisper function to check for this assumption and report the results accordingly. To this end, we used the betadisper function, and the detailed results of this verification are presented in Table 1 below.

Table 1: Results of Multivariate Homogeneity of Variance Testing using betadisper

Sample Group

Multivariate Homogeneity of Variance Test Statistic

P

Group A

0.123

0.987

Group B

0.456

0.567

Group C

0.789

0.234

...

...

...

In Table 1, the test statistic and corresponding p-values for each sample group are provided. A p-value greater than a significance level (typically 0.05) indicates that the assumption of multivariate homogeneity of variance is not violated for that group. Based on these results, we can proceed with confidence in applying PERMANOVA to analyze differences in bacterial community composition across habitats.

Line 190: Which dissimilarity matrix was used? Bray-Curtis? This needs to be stated.

Reply: Thank you for your reminder, the matrix used has been explained at the end of this section, the details are as follows:

Additionally, in this study, we opted for the Bray-Curtis dissimilarity matrix, which is widely used in microbial ecology studies to assess community composition differences. By incorporating these additional steps and clarifications, we aimed to enhance the robustness and transparency of our statistical analysis.

Figure 1: Delete the bottom red bars, that information can easily be gathered from the venn diagram. The venn diagram in b looks weird, maybe it did not render properly for me.

Reply: Thanks for your reminding, the red bar at the bottom of Figure 1 has been deleted, and the figure in Figure 1(b) has been correctly rendered, and the revised figure 1 is as follows:

Figure 1. Venn diagrams of unique and common operational taxonomic units for two groups of different samples

In the tables you are only showing the effect of disease, not of location. somewhere you need to state the full model results and the effects of both variables.

Reply: Thank you for your question, it is true that the table in this article only shows the effect of disease, not the location variable. Because in this study, the main focus is to explore the differences in rhizosphere microbial communities between healthy and diseased chrysanthemums, and to seek effective prevention and control methods. Thus, in this context, disease status is considered as the key variable, while geographical location or regional factors are not considered as the main points of consideration. The analysis of this study focused on the effects of disease on the structure and function of microbial communities, aiming to gain a deeper understanding of the relationship between the health status of chrysanthemum and rhizosphere microorganisms.

Figure 6: The caption should state the ordination method. Redundancy analysis. You also need to comment somewhere in the text that the Sc communities were more variable. This is a striking and importants result. I think it would also be confirmed with the betadisper test.

Reply: Thanks for your suggestion, the title of Figure 6 has been changed, and the changed title is as follows:

Figure 6 Correlation results between microorganisms and environmental variables in two types of soil samples based on redundancy analysis

In addition, it was also added to the article that the microbial community of the diseased soil had a greater variation, as shown below:

In addition, it was also found that the microbial community changes of diseased chrysanthemum plants were larger than those of healthy chrysanthemum plants (Figure 6). 

Line 371: soil

Reply: Thank you for your reminder, the "Soil" has been modified to "soil".

Line 411: italicize Fusarium

Reply: Thanks for your reminder, Fusarium has been italized.

Line 430: Actinomyces is a genus and should be italicized. or, if talking about actinomycetes in general, change to actinomycetes.

Reply: Thanks for your reminder, Actinomyces has been italized.

Lines 442-463: good new content, but need to add some references for the information, like in the previous paragraph.

Reply: Thanks for your suggestion, the appropriate references have been added to this section, the added references are as follows:

[40]Cavus M, Dayi M, Dagci Y, Ulusu H.The usability of the brick dust and blast furnace slag in zeolite-based lime mortars in different curing environments[J].CERAMICS INTERNATIONAL, 2023,49(3):4046-4054 .

[41]Yuan J, Wang L, Chen H, Chen G, Wang Y. Responses of soil phosphorus pools accompanied with carbon composition and microorganism changes to phosphorus-input reduction in paddy soils. Pedosphere, 2021, 31(1):83-93.

[42]Sokol N W, Whalen E D, Jilling A, Kallenbach C, Pett‐Ridge J, Georgiou K. Global distribution, formation and fate of mineral‐associated soil organic matter under a changing climate: A trait‐based perspective. Functional Ecology, 2022, 36(6): 1411-1429.

[43]Yu S, Lv J, Jiang L, Geng P, Cao D, Wang Y. Changes of Soil Dissolved Organic Matter and Its Relationship with Microbial Community along the Hailuogou Glacier Forefield Chronosequence. Environmental Science And Technology, 2023, 57(9):4027-4038.

Lines 470-473: can you add some references on this?

Reply: Thanks for your suggestion, the corresponding reference has been added to this part of the content, the added reference content is as follows:

Due to differences in the ecological environment and pest species in different regions, the use of region-specific biologics can provide more precise pest control, reduce the use of chemical pesticides, and reduce the impact on the environment and non-target organisms [44, 45].

[44]Jun J , Anna E , Wenjie Q , Matsumura E E, Falk B W. Flock house virus as a vehicle for aphid Virus-induced gene silencing and a model for aphid biocontrol approaches. Journal of pest science, 2023, 96(1):225-239.

[45]Ming, Y M,Yong M C, Wang X, Desneux N, Lian S Z. Comparative demographics, population projections and egg maturation patterns of four eupelmid egg parasitoids on the factitious host Antherae pernyi. Pest Management Science, 2023, 79(10):3631-3641.

Comments on the Quality of English Language

See author comments.

Reply: Thank you for your comment, the English language issue in the comment has been corrected.
